# Imaging biological tissue with high-throughput single-pixel compressive holography

Daixuan Wu [1,2], Jiawei Luo[1,2], Guoqiang Huang[1,2], Yuanhua Feng[3], Xiaohua Feng[4], Runsen Zhang[1,2,5], Yuecheng Shen [1,2✉] & Zhaohui Li [1,2,6✉]

Single-pixel holography (SPH) is capable of generating holographic images with rich spatial information by employing only a single-pixel detector. Thanks to the relatively low dark-noise production, high sensitivity, large bandwidth, and cheap price of single-pixel detectors in comparison to pixel-array detectors, SPH is becoming an attractive imaging modality at wavelengths where pixel-array detectors are not available or prohibitively expensive. In this work, we develop a high-throughput single-pixel compressive holography with a space-bandwidth-*time* product (SBP-T) of 41,667 pixels/s, realized by enabling phase stepping naturally in time and abandoning the need for phase-encoded illumination. This holographic system is scalable to provide either a large field of view (~83 mm$^2$) or a high resolution (5.80 μm × 4.31 μm). In particular, high-resolution holographic images of biological tissues are presented, exhibiting rich contrast in both amplitude and phase. This work is an important step towards multi-spectrum imaging using a single-pixel detector in biophotonics.

[1] Key Laboratory of Optoelectronic Materials and Technologies, School of Electronics and Information Technology, Sun Yat-sen University, Guangzhou, China. [2] Guangdong Provincial Key Labratory of Optoelectronic Information Processing Chips and Systems, School of Electronics and Information Technology, Sun Yat-sen University, Guangzhou, China. [3] Department of Electronic Engineering, College of Information Science and Technology, Jinan University, Guangzhou, China. [4] Department of Bioengineering, University of California, Los Angeles, USA. [5] Institute of Photonics Technology, Jinan University, Guangzhou, China. [6] Southern Marine Science and Engineering Guangdong Laboratory (Zhuhai), Zhuhai, China. ✉email: shenyuecheng@mail.sysu.edu.cn; lzhh88@mail.sysu.edu.cn

Pixel-array detectors, such as CCD and CMOS cameras, were commonly used in the traditional imaging scheme. However, these detectors are only cost-effective and maintain good performance within a certain spectrum range. In contrast to pixel-array detectors, single-pixel detectors have lower dark-noise production, higher sensitivity, faster response time, and a much cheaper price. Moreover, they have been demonstrated with great performance across almost the entire spectrum range. Therefore, single-pixel imaging (SPI), an emerging computational method that employs a single-pixel detector at the receiving end, offers great potential for optical imaging at wavelengths where pixel-array detectors are not available or prohibitively expensive. Enabled by this property, SPI has been demonstrated with great success when operating with infrared light[1], Terahertz wave[2], and even photoacoustic signal[3]. Instead of acquiring spatial information through parallel detection, SPI relies on using a spatial light modulator (SLM) to display a series of ordered patterns and then computationally reconstruct spatial information from a series of measurements. Without compressive sensing, the number of effective pixels in the reconstructed image equals the number of ordered patterns being displayed. Since the first demonstration called flying-spot camera in 1884 by Nipkow et al.[4,5], SPI has been later demonstrated with advantages when imaging through scattering media[6–8] or in scarce illumination with compressive sensing[9,10]. By employing various coding mechanisms including Hadamard bases[8,11–23], Fourier bases[12,24–26], and random patterns[27], SPI has also been extended and demonstrated with applications in full-color imaging[13], multispectral imaging[14], time-resolved imaging[15], and three-dimensional imaging[28–30]. Recently, adaptive and smart sensing with dynamic supersampling was reported to combine with compressive sensing in SPI. The enabling feature of this approach is to rapidly record fast-changing features by dynamically adapting to the evolution of the scene. Thus, it significantly shortens acquisition time without considerably sacrificing spatial information[31].

From the perspective of optical imaging, the successful modeling of complex-valued biological samples with both amplitude and phase information is of great significance in biophotonics. For example, many thin biological tissues exhibit low scattering and absorption when interacting with light, leading to low contrast when being directly imaged without staining process using a conventional microscope. Even for relatively thick tissues whose amplitude images can provide enough contrast, their corresponding phase images always serve as a good supplement. Borrowing the concept of the Shack–Hartmann sensor, a non-interferometric phase image was demonstrated using a single-pixel detector[32]. However, the special requirement of using a lateral position detector sacrifices the advantages that conventional SPH holds. Since the fast oscillation of the diffracted light prohibits direct measurement of phase information using modern optical detectors, there is a strong desire to develop an efficacious imaging modality that can provide complex-valued images to study the microscopic structures of a myriad of biological tissues[33–37]. This capability can also benefit a variety of applications in adaptive optics[38,39], surface contour[40,41], wavefront sensing[42,43], optical metrology[44–46], and ultrafast optics[47–49].

To beat down the fast oscillation of the diffracted light to the regime that modern detectors can reach, holographic approaches that employ additional reference beams become one of the most effective and intuitive ways to retrieve field information[50–52]. Thus, when combined with this approach, SPI can be further generalized to extract complex-valued information from the sample, naming single-pixel holography (SPH). Back in 2013, Clemente et al. developed a framework for using a liquid-crystal-based SLM and a bucket single pixel to image phase objects[16].

Later, digital micromirror devices (DMDs) were employed as the dominant devices to increase the illumination speed. With DMDs, fast fluorescence imaging and phase imaging were simultaneously realized in a compact SPH system[20]. People also explored several modifications to boost the performance of SPH, including choosing appropriate orders of various illumination patterns for compressive sensing[12,53] and developing common-path interference for robustness[18–21].

Based on the review articles covering this topic[54,55], Table 1 summarizes the performance of some representative works about SPI[8,12–15,24] and SPH[16–23,25,26] reported in the literature. As systems of SPI were developed in the early years, liquid-crystal-based SLMs were usually employed. Although being slow in the refresh rate, these SLMs can simultaneously modulate lots of pixels independently, allowing $256 \times 256 = 65,536$ pixels in the reconstructed images to be frequently obtained[12]. Moreover, these systems were generally designed to image macroscopic objects with a relatively large field of view (FOV). As a comparison, systems of SPH systems prefer using DMDs with fast refresh rates[20–23,25,26]. However, DMDs support only binary-amplitude modulation so that phase stepping inherent in holography has to be realized at the cost of available pixels through the Lee hologram[56] or superpixel method[57]. As a result, it is conspicuous that the number of pixels in the reconstructed images with SPH (the largest one is $128 \times 128 = 16,384$[16,20,21]) is normally smaller than that with SPI. For a convenient and straightforward comparison of SPH systems, some key parameters were highlighted with asterisks. For example, the best lateral resolution was reported by Shin et al. as $0.4\,\mu m \times 0.4\,\mu m$[21], while the largest FOV was reported by Hu et al. as $11.7\,mm \times 11.7\,mm$[25]. Since there always exists a trade-off among imaging speed, FOV, and lateral resolution, it is inappropriate to compare the performance according to only one performance index. A figure-of-merit parameter, defined as the space-bandwidth-*time* product (SBP-T)[58,59], can be used for a fair comparison of SPI/SPH, which represents the throughput of the system. Mathematically, SBP-T is computed as the number of pixels in the reconstructed image divided by the total acquisition time consumed, accounting for the information collected per unit time. For SPH, an additional factor of 2 is multiplied in the consideration for both the amplitude and the phase. By going through Table 1, the largest SBP-T that has been achieved with SPH before this work was 14,667[26]. We note here that compressive sensing, including adaptive and smart sensing, is effective in breaking the spatial and temporal trade-off for most SPI and SPH systems. However, the improvement in SBP-T is ambiguous to be quantified, especially when the target sample is not specified. Therefore, compressive sensing was not considered when estimating the parameters displayed in Table 1 for a fair comparison.

Two major factors limit the throughput of SPH in current practice: (1) the phase stepping inherent in holography required a few patterns being displayed for each order, thus inevitably slowing down the imaging process; (2) both the Lee hologram and the superpixel method were realized at the cost of independent pixels, therefore reducing the number of effective pixels in the reconstructed image.

In this work, we overturn this practice by developing high-throughput SPH. Instead of actively performing phase shifting, a beat frequency is introduced between the signal beam and the reference beam, thereby realizing phase stepping naturally in time by exploiting the framework of heterodyne holography. Moreover, instead of generating phase patterns with DMDs by scarifying pixels, we show both theoretically and experimentally that non-orthogonal binary-amplitude Hadamard patterns can be used for holographic reconstruction as well. Thus, by directly

**Table. 1 List of the parameters in both single-pixel imaging and single-pixel holography.**

| | Ref. | Pixel size (µm) | Refresh time (ms) | Number of pixels | Measurements | Total acquisition time (s) | Resolution (µm) | FOV (mm) | SBP-T (pixels/s) | Biological sample? |
|---|---|---|---|---|---|---|---|---|---|---|
| Single-pixel imaging | 13 | NA | 1.54 | 256 × 128 = 32,768 | 32768 | 50.46 | ≈780 | 200 × 100 | 650 | No |
| | 24 | NA | 150 | 245 × 245 = 60,025 | 120,052 | 18,007.8 | ≈653 | 160 | 3.3 | No |
| | 14 | 13.68 | 20 | 64 × 64 = 4096 | 4096 | 81.9 | ≈703 | 45 | 50 | No |
| | 12 | 13.68 | 0.5 | 256 × 256 = 65,536 | 131,072 | 65.6 | ≈781 | ≈200 | 1000 | No |
| | 12 (large FOV) | 5.4 | 200 | 256 × 256 = 65,536 | 131,072 | 26,216 | 5.47 | 1.4 | 2.5 | No |
| | 12 (high resolution) | | | | | | | | | |
| Single-pixel holography | 15 | 13.68 | 1000 | 64 × 64 = 4096 | 4096 | 4096 | NA | NA | 1 | No |
| | 8 | NA | 0.05 | 32 × 32 = 1024 | 1024 | 0.0512 | NA | NA | 20,000 | No |
| | 16 | 19 | ≈13.89 | 128 × 128 = 16,384* | 16,384 | ≈227.6 | 19 | 2.4 | 144 | No |
| | 17 | 20 | ≈16.67 | 64 × 64 = 4096 | 4096 | ≈68.3 | 160 | 10.2 | 120 | No |
| | 22 | 13.68 | ≈0.0455 | 64 × 64 = 4096 | 16,384 | ≈0.7447 | ≈52.5 | 3.4 | 11,000 | No |
| | 23 | 13.68 | 0.5 | 64 × 64 = 4096 | 16,384 | 8.192 | ≈52.5 | 3.4 | 1000 | Yes* |
| | 20 | 13.68 | 0.05 | 128 × 128 = 16,384* | 49,152 | 2.5 | ≈82 | ≈10.5 | 13,333.4 | No |
| | 20 (large FOV) | 13.68 | 0.05 | 128 × 128 = 16,384* | 49,152 | 2.5 | ≈12.5 | ≈1.6 | 13,333.4 | No |
| | 20 (high resolution) | | | | | | | | | |
| | 21 | 13.68 | 0.1 | 128 × 128 = 16,384* | 49,152 | 4.9 | ≈0.4* | ≈0.052 | 6666.8 | No |
| | 18 | 20 | 100 | 16 × 16 = 256 | 1024 | 102.4 | 476.3 | 7.62 | 5 | No |
| | 25 | 10.8 | 0.0562 | 81 × 81 = 6561 | 26,244 | 1.5 | 144 | 11.7* | 8896.8 | No |
| | 19 | 13.68 | 0.0455 | 64 × 64 = 4096 | 16,384 | ≈0.7447 | 109.5 | 7 | 11,000 | No |
| | 26 | 13.68 | 0.0455 | 103 × 103 = 10,609 | 31,827 | ≈1.448 | 68.4 | 7 | 14,666.6 | No |
| | This work (large FOV) | 13.68 | 0.048 | 256 × 256 = 65,536* | 65,536 | 3.14 | 58.0 × 43.1 | 14.9 × 11.1* | 41,666.6* | No |
| | This work (high resolution) | 13.68 | 0.048 | 256 × 256 = 65,536* | 65,536 | 3.14 | 5.80 × 4.31 | 1.49 × 1.11 | 41,666.6* | Yes* |

The key parameters that represent the most advanced indexes in SPH before and after this work were highlighted with asterisks, respectively.
NA: not applicable.

projecting the desired amplitude patterns to the sample, the developed high-throughput SPH can achieve an SBP-T of 41,667 pixels/s, which is about 3 times larger than the largest one reported in the literature[26]. It is worth noting that this value was achieved using a single-pixel detector. Moreover, the number of pixels in the reconstructed image can be up to $256 \times 256 = 65,536$, which is about 4 times larger than the largest one reported with SPH before. The developed holographic system can be adapted for different application scenarios by functioning under different operational modes. For example, we can operate under large-FOV mode (14.9 mm × 11.1 mm) to monitor the environment[12,13,24] or switch to high-resolution mode (5.80 μm × 4.31 μm) to scrutinize microstructures[21,23]. In the microscopic world, one of the most imperative tasks is to image biological tissue. Unfortunately, however, to date, applying SPI/SPH to imaging microscopic structures in biological tissue has been barely reported, mainly due to the limited performance of the imaging system and the relatively low scattering contrast of biological samples. Until very recently, González et al. employed SPH to image the wing of an insect with a lateral resolution of 52.5 μm[23]. This regrettable situation considerably hinders SPI/SPH from being widely adopted in biophotonics. Here, we bridge this gap by developing SPH to image biological tissue from mouse tails and brains, revealing rich information in both amplitude and phase. The FOV and the lateral resolution of corresponding figures are 1.51 mm × 1.11 mm and 5.80 μm × 4.31 μm, respectively. This work constitutes an important step towards future high-throughput image modalities using a single-pixel detector and shows great promise of applying SPI/SPH to image microscopic structures of biological tissue.

## Results

**Verification of holographic performance**. To examine the performance of the holographic system, a standard positive 1951-USAF resolution test target (R3L3S1P, Thorlabs) was used as the sample. Firstly, the system was operated under the large-FOV mode. Both strategies of using $768 \times 768$ pixels with $3 \times 3$ binning and $512 \times 512$ pixels with $2 \times 2$ binning are applied to verify the performance and scalability of the system. For the first binning strategy, the FOV and the lateral resolution are reported above. Thus, element 4 of group 3 (44.25-μm width) in the resolution target is the finest structure that can be identified. During experiments, 65,536 Hadamard-like patterns were sequentially displayed and the total imaging process only took 3 s (The corresponding raw data are available in public repository Zenodo[60]). Figure 1a, b shows the reconstructed complex-valued images for groups 2 to 4 of the resolution target, including both the amplitude and phase. As shown in the upper inset of Fig. 1a, the amplitude image with a smaller FOV to magnify element 4 of group 3 is captured by a conventional microscope. For visualization purposes, the corresponding one-dimensional (1D) profile denoted by the brown bracket within the resolution target is shown in the inset below, exhibiting three well-separated narrow dips (vertical lines), one wide dip (horizontal line), and one narrow dip (bottom area of "4"). This result demonstrates that the resolution achieved by using our system meets theoretical expectations. Although the contrast of the phase imaging is not as good as its amplitude counterpart possibly due to insufficient signal-to-noise ratio, similar structures belong to element 4 of group 3 can be observed as well. Corresponding 1D profile of phase image at the same position is also provided in the inset. Besides, regarding the second binning strategy, the FOV and lateral resolution minify to 9.91 mm × 7.37 mm and 38.7 μm × 28.8 μm, and corresponding holographic images were provided in the Supplementary Note 7.

To further facilitate the imaging process, compressive sensing can be used. To be consistent with the general significance of natural scenes, several orderings of Hadamard bases have been demonstrated previously. Depending on the uniqueness of each sample, these orderings exhibit slightly different performances[53,61]. In this work, an ordering of square sampling path (detailed in the Supplementary Note 5) was applied for sampling, and a direct inverse fast Hadamard transformation using the same set of bases was adopted for reconstruction. Thus, incoherence is confirmed between the bases employed for the measurement and the reconstruction, which is essential to make compressive sensing work for our holographic system[62]. Figure 1c, d shows the reconstructed amplitude and phase images using different sampling ratios (SRs), defined as the ratio between the number of measurements used for reconstruction and the number of pixels in the reconstructed image. For the series of amplitude images in Fig. 1c, most structures in group 3 are still identifiable, even when the SR was decreased to 6.25%. Moreover, since a smaller SR is usually accompanied by a higher contrast-to-noise ratio, the amplitude image reconstructed with the SR of 12.5% looks even better than those reconstructed with the SR of 50%. We note that this observation is because information carried by Hadamard-like patterns with higher orders is more sensitive to measurement noises. A quantitative study on how measurement noises affect the image quality of the reconstructed image under different choices of SRs can be found in the Supplementary Note 6. Thus, the SR of 25% might be a suitable choice for the noise level in current experimental conditions (the standard deviation of the measurement noise is around 0.1% of the measured value). However, when the SR was further reduced to 3.125%, artifacts emerge so that no clear lines can be identified. Since the square path we employed for compressive sensing follows the order from low spatial frequency to high spatial frequency, the spatial resolution is expected to degrade along with the square root of the SR. Nonetheless, this observation confirms that by properly selecting the SR, compressive sensing can greatly shorten the acquisition time without sacrificing too much imaging quality. The series of phase images reconstructed with various SRs are shown in Fig. 1d. As long as the SR ≥ 12.5%, the quality of the phase images is still acceptable.

**Holographic imaging of stained tissue from mouse tails**. Having demonstrated operating at large-FOV mode, we then switched to the high-resolution mode. The strategy of using $768 \times 768$ pixels with $3 \times 3$ binning is also applied to guarantee light delivery. It is estimated that the FOV and the lateral resolution are 1.49 mm × 1.11 mm and 5.80 μm × 4.31 μm, respectively, which make this holographic system well suited for imaging biological samples. We then examined its performance by imaging the resolution target, as shown in Fig. 2(a). For visualization purposes, 1D profile of element 6 of group 6 (4.386-μm width, denoted with a black bracket) is plotted in the lower inset. One wide dip and three separated narrow dips can be identified, which is consistent with the same structure measured by a conventional microscope (the upper inset). Then, we proceed to image biological tissue. Figure 2(b) shows the image of a 10-μm-thick slice of stained tissue from mouse tails, captured using a bright-field microscope. In this image, several different types of tissue such as muscle, cortical bone, and cancellous bone are marked. Figure 2c–e shows a series of reconstructed holographic images for different parts of the stained tissue, indicated by three labeled diamond-shaped boxes in Fig. 2b. For each part, 65,536 Hadamard patterns were sequentially displayed and the data acquisition took about 3 s (The corresponding raw data are available in public repository Zenodo[60]). It is conspicuous that all

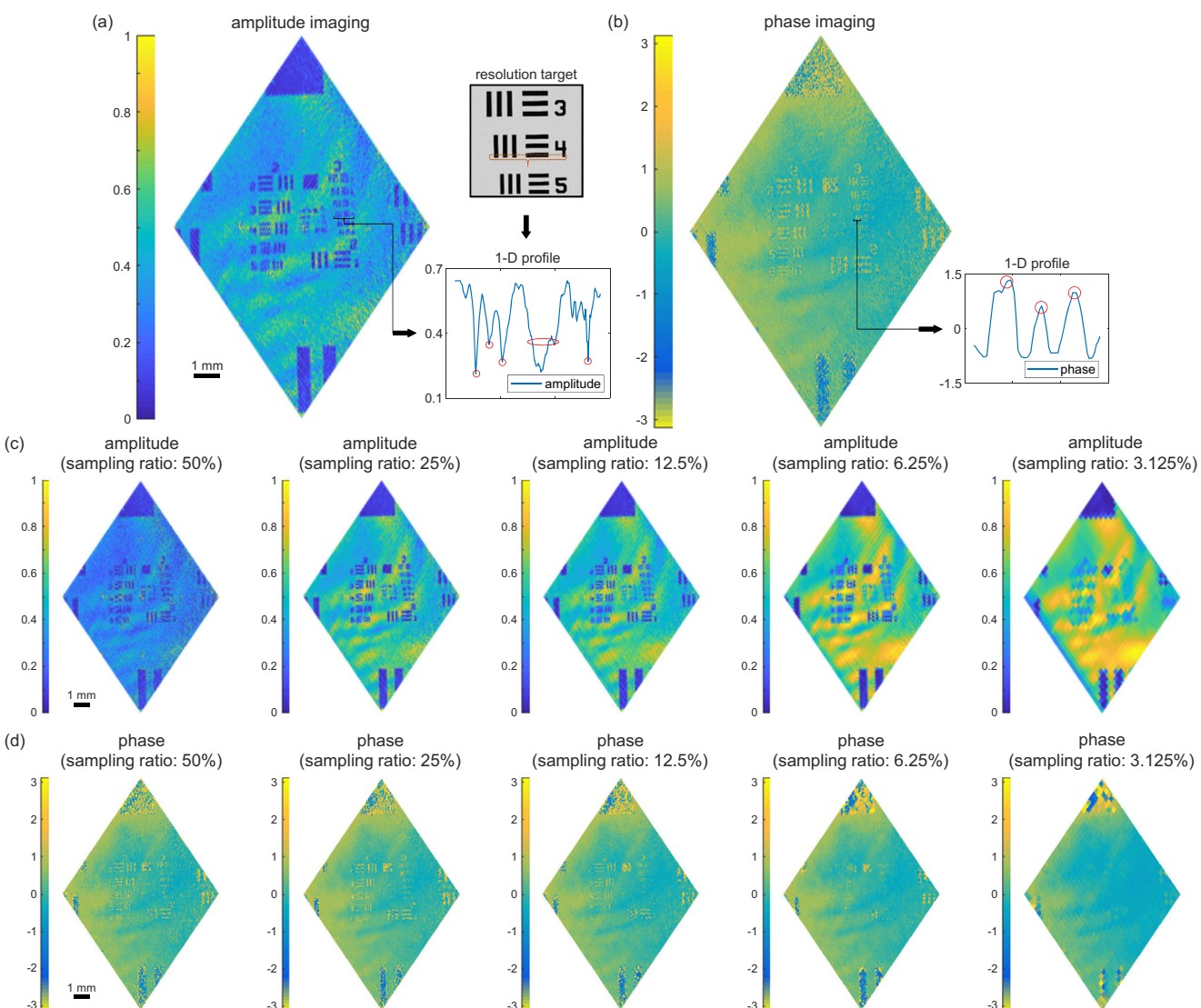

**Fig. 1 Performance of the high-throughput SPH in large-FOV mode.** A standard positive 1951-USAF resolution test target (R3L3S1P, Thorlabs) was used as a testing sample. 3 × 3 pixels binning strategy was adopted for 768 × 768 pixels, leading to 256 × 256 superpixels. **a** Reconstructed amplitude image of the resolution target. The upper inset: the image captured by conventional microscopy; the lower inset: the corresponding 1D profile of element 4 of group 3. **b** Reconstructed phase image of the resolution test target. The corresponding 1D profile is shown in the right inset. **c**, **d** The amplitude and wrapped phase images are reconstructed with different SRs of 50%, 25%, 12.5%, 6.25%, 3.125%. The corresponding scale bar is 1 mm.

three amplitude images show lots of intricate details and are in good agreement with the one in Fig. 2(b), manifesting great distinctions among different types of tissue. For stained tissue, the reconstructed phase images are analogous to their amplitude counterparts. Another piece of stained tissue from mouse tails with the same thickness was also imaged using the holographic system. Corresponding holographic images can be found in the Supplementary Note 9. These demonstrations show that the developed high-throughput SPH is capable of revealing delicate microscopic structures of biological tissue.

We also applied compressive sensing for these images. As a typical example, we targeted the area that corresponds to that in Fig. 2e. The reconstructed amplitude and phase images using various SRs are illustrated in Fig. 3a, b, respectively. In general, for both amplitude and phase images, the smaller the SR, the less detailed structures can be visualized. These images demonstrate the effectiveness of compressive sensing with small SRs when dealing with biological samples that contain rich detailed information.

**Holographic imaging of unstained tissue from mouse brains.** Having demonstrated holographic imaging of stained tissue from mouse tails, we then switched to image unstained tissue from mouse brains. Generally, images of unstained tissue exhibit low contrast in amplitude, but providing sufficient contrast through phase imaging. Before proceeding, we quantified the performance of the holographic system in phase by imaging a quantitative phase resolution target (QPT, BenchMark Tec). Figure 4a shows the reconstructed phase image of the resolution target, containing groups 6 and 7. To quantify the resolving power in phase, the corresponding 1D profile denoted by the black bracket within the resolution target, crossing element 6 of group 6 (4.386-μm width), is shown in the upper inset. Three verticle bars can be differentiated, indicating that the theoretical resolution under the high-resolution mode (5.80 μm × 4.31 μm) was achieved in phase. We further quantified the accuracy of the measurement in phase by estimating the phase difference between the bar enclosed by a dashed box at the bottom of the target and its adjacent background. 1D profile that crosses both the bar and the background

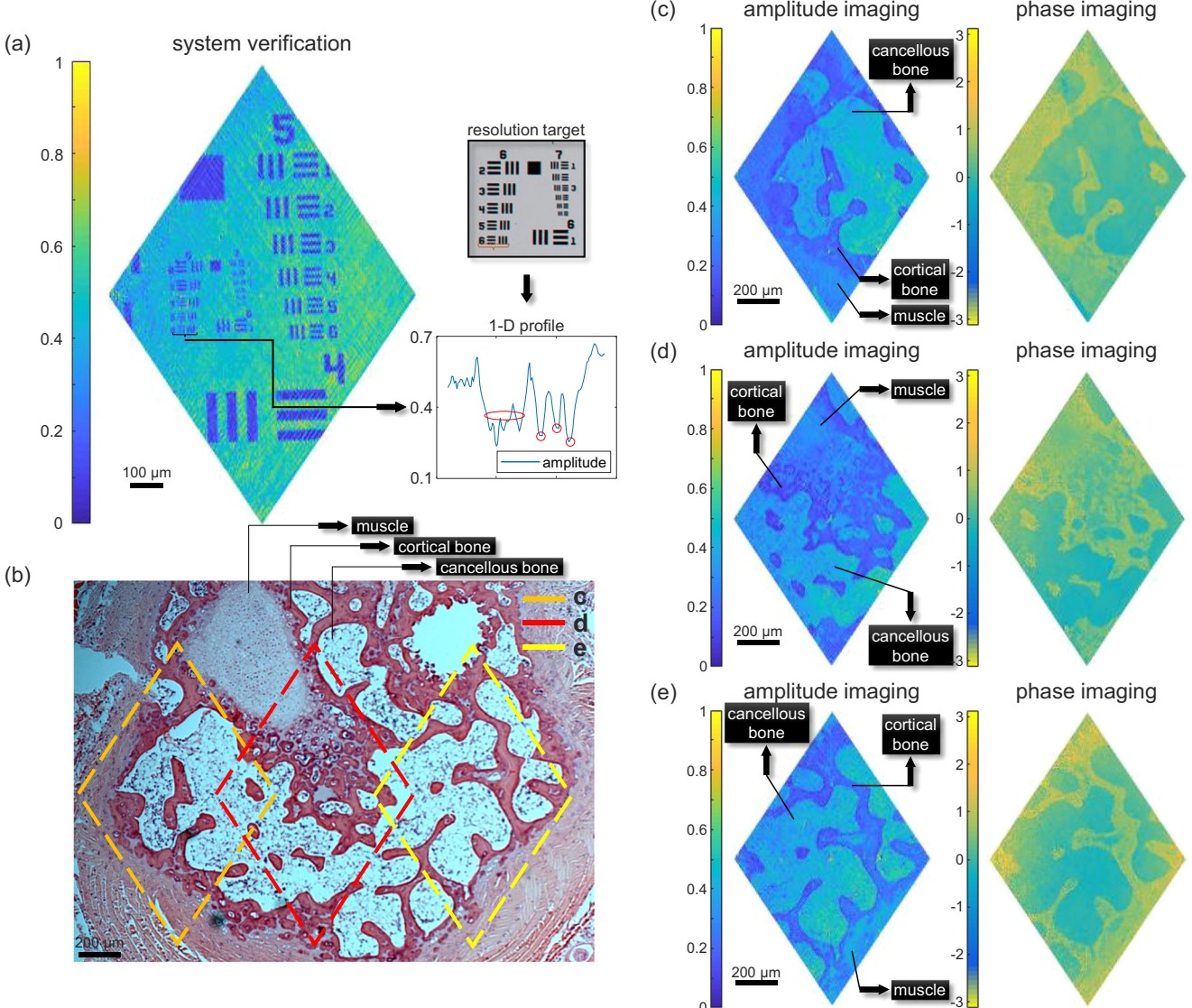

**Fig. 2 Performance of the high-throughput SPH with stained tissue from mouse tails in high-resolution mode. a** Reconstructed amplitude image of a resolution target. The upper inset: the image of the resolution target (containing groups 6 and 7) captured by a conventional microscope; the lower inset: the corresponding 1D profile of element 6 of group 6. **b** The image of a slice of stained tissue from mouse tails, captured using a conventional microscope. Three diamond-shaped boxes represent the area being measured by the holographic system. **c–e** The reconstructed amplitude and phase images for different parts of the stained tissue. The corresponding scale bar is 200 μm.

is shown in the lower inset using blue circles, exhibiting a stepped structure. The averaged phase difference was estimated to be $\triangle\varphi_{exp} \approx 1.691$ rads, yielding a phase error of only 0.104 rads ($\leq \lambda/60$) compared to the actual phase difference ($\triangle\varphi = 1.795$ rads). Detailed analysis on imaging the quantitative phase resolution target can be found in the Supplementary Note 8. Then, we proceed to image unstained biological tissue. Figure 4b shows the image of an 80-μm-thick slice of unstained tissue from mouse brains, captured using a bright-field microscope. In this image, some representative types of tissue were denoted with black arrows, such as piriform cortex, perirhinal cortex, olivary pretectal nucleus, and white matter. For unstained tissue, it is usually challenging to identify different structures in amplitude, due to the insufficient contrast in transmission. Figure 4c–e shows a series of reconstructed holographic images for different parts of the mouse brain, indicated by three labeled diamond-shaped

boxes in Fig. 4b. Same as the one captured using the conventional microscope, these amplitude images only manifest rough outlines with low contrast. By comparison, phase images show rich details that can be hardly seen in their amplitude counterparts. For example, the piriform cortex and the perirhinal cortex are identifiable in phase images. Data used for producing Fig. 4c–e are available in public repository Zenodo[60]. These results demonstrate the capability of employing SPH to image relatively transparent biological tissue.

For the selected area in Fig. 4c, a series of holographic images reconstructed with compressive sensing at various SRs are illustrated in Fig. 5 as well. Similar to before, these images show the effectiveness of compressive sensing in dealing with biological tissue that contains rich information in phase. We also applied SPH to image other pieces of unstained tissues from mouse brains with different thicknesses, ranging from 10 to 120 μm, with details shown in the Supplementary Note 10.

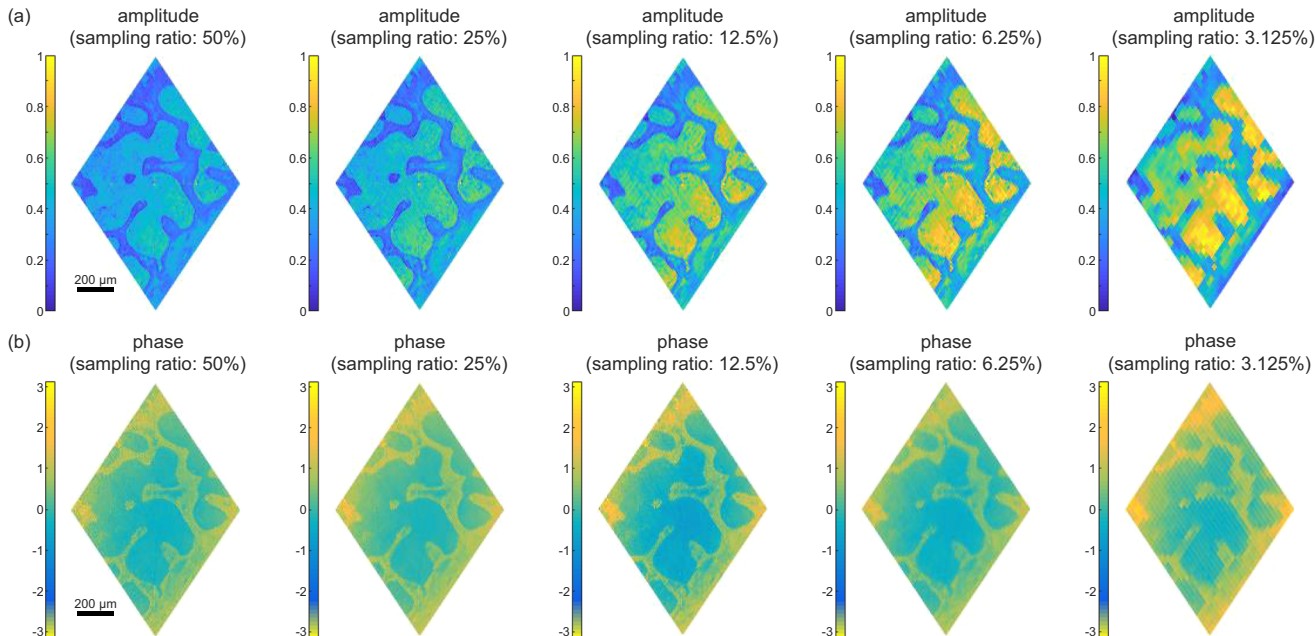

**Fig. 3 Reconstruction of holographic images for the piece of mouse tail with compressive sensing.** The amplitude and wrapped phase images are reconstructed with different SRs of 50%, 25%, 12.5%, 6.25%, 3.125%. The corresponding scale bar is 200 µm.

## Discussion

In this work, the highest Hadamard patterns being displayed is 65,536, restricting the number of pixels in the reconstructed image to be $256 \times 256$. Such a crucial factor is limited by the finite onboard memory of the DMD (64 Gbits), leading to insufficient usage of its $1024 \times 768$ independent micromirrors. Therefore, expanding the onboard memory of the DMD is the key to further increase the number of pixels in the reconstructed image. Moreover, given finite laser power and inevitable energy loss of the holographic system, we noticed that light delivered to the sample is not strong and the measured signal contrast is just around the edge when retrieving coefficients for high orders. Taking the high-resolution mode as an example, the light intensity being projected onto the biological tissue is only 6 mW/cm² , which is about 33 times below the American National Standards Institute (ANSI) safety limit (200 mW/cm²). These values also explain why a $3 \times 3$ binning was adopted for $768 \times 768$ pixels to boost light delivery to biological tissue during the experiments. In practice, when loading patterns with high spatial frequencies, not all light can be collected by the photodetector if the sensing area is not large enough. This phenomenon can degrade the quality of the reconstructed images in SPI, especially for fine details. Nonetheless, it is surprising to find that SPH is immune to this effect. This observation is because only the interference term with zero spatial frequency contributes to the reconstruction process of SPH. Therefore, future works will focus on increasing the laser power and minimizing energy loss in the holographic system, without the need to enlarge the sensor area of the detector. Due to the huge computational burden of imaging reconstruction, real-time imaging of live cells with time-varying features is still unachievable with the current holographic system. We envision that this constraint can be potentially alleviated through the assistance of deep learning and compressive sensing[63]. Besides, although only two operational modes, i.e., the large-FOV mode and the high-resolution mode, were demonstrated, the imaging parameters of this single-pixel holographic system are scalable to other values, either by choosing different pairs of lenses in the 4*f* system (physically) or using different pixel binning strategy of the DMD (digitally).

In conclusion, we developed high-throughput SPH and imaged biological tissue with high resolution. To realize phase stepping, we creatively introduced heterodyne holography into SPH by using two acousto-optic modulators (AOMs) with slightly different modulation frequencies, which increases the amount of information collected per second. For many imaging indicators, such as FOV, lateral resolution, number of pixels, and SBP-T, the developed holographic system here remain advanced among the existing systems of SPH (see Table 1). Moreover, the throughput of our system reaches an SBP-T of 41,667 pixels/s, thus being at least 3 times larger than the largest one reported in the literature[26]. Experimentally, both the large-FOV mode and the high-resolution mode of this holographic system were demonstrated by imaging resolution targets and biological tissues. In particular, biological tissues including stained slices from mouse tails and unstained slices from mouse brains were imaged under the high-resolution mode. Different types of tissues can be identified in the reconstructed amplitude and phase images, showing that SPH is well suited for biomedical applications. We envision that the developed high-throughput SPH is promising to promote multi-spectrum imaging by providing high-resolution complex-valued images for a variety of biological tissues within a broad spectrum range.

## Methods

**Principle of high-throughput SPH.** We first describe the operating principle of the high-throughput SPH. Mathematically, the sample to be imaged is described by a complex function $O\left(\vec{r}\right) = A\left(\vec{r}\right) e^{i\phi\left(\vec{r}\right)}$, where $A\left(\vec{r}\right)$ represents the spatially-varying transmissivity and $\phi\left(\vec{r}\right)$ denotes the accumulated phase during light propagation. SPI relies on the fact that $O\left(\vec{r}\right)$ can be decomposed using a set of orthogonal Hadamard bases $H_n\left(\vec{r}\right)$ that only contains the values of "+1" and "−1"

$$O\left(\vec{r}\right) = A\left(\vec{r}\right) e^{i\phi\left(\vec{r}\right)} = \frac{1}{\sqrt{N}} \sum_n a_n e^{i\varphi_n} H_n\left(\vec{r}\right) \qquad (1)$$

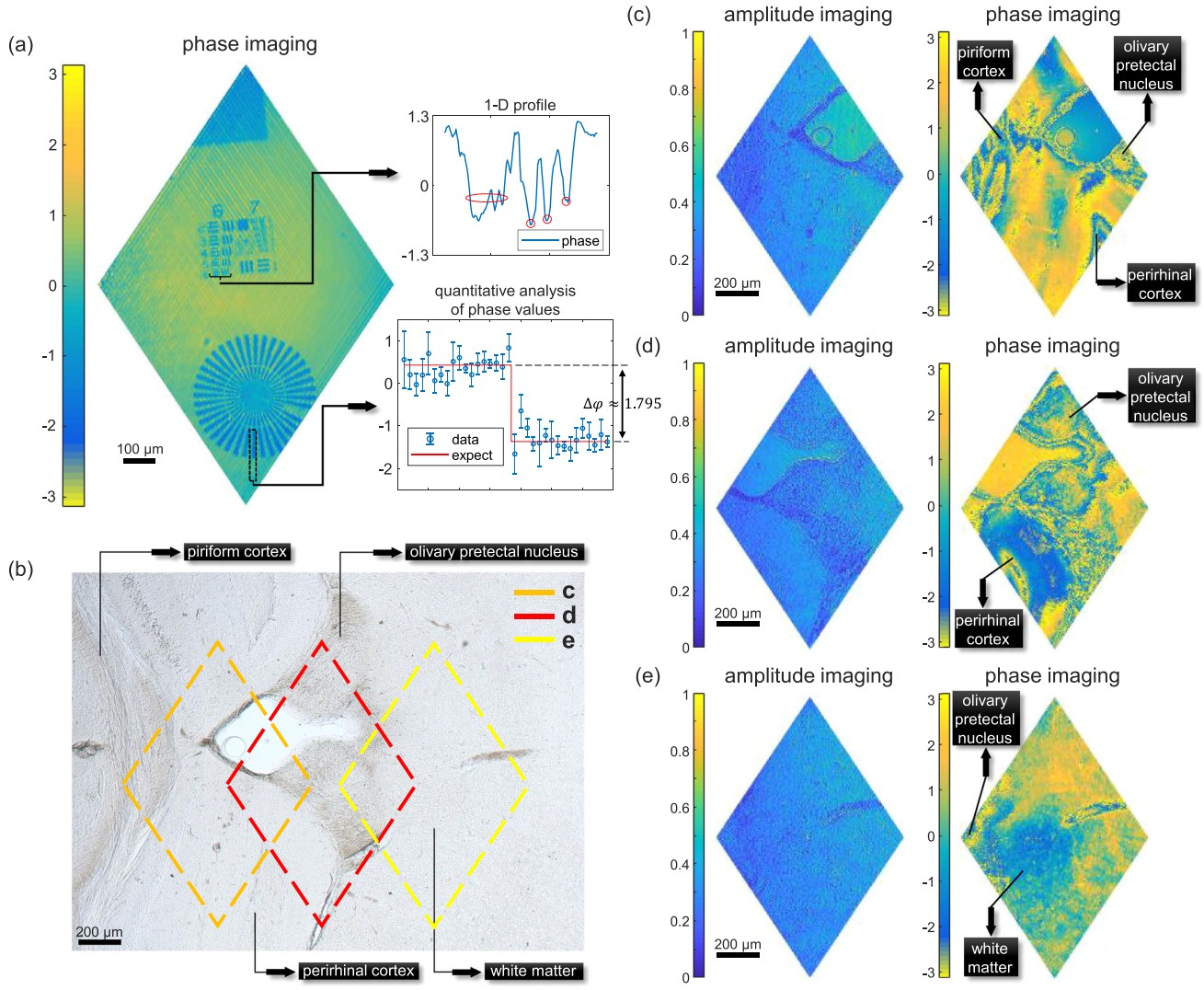

**Fig. 4 Performance of the high-throughput SPH with unstained tissue from mouse brains in high-resolution mode. a** Reconstructed phase image of a quantitative phase target. The upper inset: the corresponding 1D profile of element 6 of group 6; the lower inset: quantitative analysis on the accuracy of the reconstructed phase values (actual phase difference $\triangle\varphi = 1.795$ rads). The data are presented as mean values and standard deviations, resulting in a phase error of about 0.104 rads (from $n = 4$ independent measurements). **b** The image of a slice of unstained tissue from mouse brains, captured using a conventional microscope. Three diamond-shaped boxes represent the area being imaged by the holographic system. **c–e** The reconstructed amplitude and phase images for different parts of the unstained tissue. The corresponding scale bar is 200 μm.

where $1/\sqrt{N}$ is the normalization constant. By exploiting the orthogonality of Hadamard bases, the complex-valued coefficient $a_n e^{i\varphi_n}$ can be obtained by taking the spatial summation of all the transmitted field over the sample surface $S$ when $H_n\left(\vec{r}\right)$ is projected:

$$a_n e^{i\varphi_n} = \frac{1}{\sqrt{N}} \int_S^1 O\left(\vec{r}\right) H_n\left(\vec{r}\right) dS \qquad (2)$$

Equation (2) indicates that the successful operation of SPH requires sequentially projecting Hadamard-encoded illumination and measuring the corresponding transmitted field on the single pixel. However, DMDs with fast refresh rates can only generate Hadamard-like patterns $\widetilde{H}_n\left(\vec{r}\right)$ with the values of "+1" and "0". Since a series of $\widetilde{H}_n\left(\vec{r}\right)$ do not form orthogonal bases, the mathematical relationships in Eqs. (1) and (2) do not hold. To modulate phase using DMDs, previous works employed either Lee hologram or superpixel method at the cost of available pixels[20–23], thereby limiting the throughput. Fortunately, we show in the following that generating phase patterns is not necessary for SPH. As shown in Fig. 6, $H_n\left(\vec{r}\right)$ and $\widetilde{H}_n\left(\vec{r}\right)$ are closely related through $H_n\left(\vec{r}\right) = 2\widetilde{H}_n\left(\vec{r}\right) - \widetilde{H}_1\left(\vec{r}\right)$ or $\widetilde{H}_n\left(\vec{r}\right) = \left(H_n\left(\vec{r}\right) - H_1\left(\vec{r}\right)\right)/2$. Here, $\widetilde{H}_1\left(\vec{r}\right)$ is the first order of Hadamard-like patterns, which refers to a term of direct current. Thus, Eq. (2) can

be rewritten as

$$a_n e^{i\varphi_n} = \frac{2}{\sqrt{N}} \int_S^1 O\left(\vec{r}\right) \widetilde{H}_n\left(\vec{r}\right) dS - \frac{1}{\sqrt{N}} \int_S^1 O\left(\vec{r}\right) \widetilde{H}_1\left(\vec{r}\right) dS$$
$$= 2\widetilde{a}_n e^{i\widetilde{\varphi}_n} - \widetilde{a}_1 e^{i\widetilde{\varphi}_1} \qquad (3)$$

Here, $\widetilde{a}_n e^{i\widetilde{\varphi}_n}$ is the spatial summation of all the transmitted field over the sample $S$ when $\widetilde{H}_n\left(\vec{r}\right)$ is projected. This equation indicates that the complex-valued coefficient $a_n e^{i\varphi_n}$ can be determined by sequentially displaying Hadamard-like patterns $\widetilde{H}_n\left(\vec{r}\right)$ as well, without the need to illuminate Hadamard bases $H_n\left(\vec{r}\right)$. To measure the transmitted complex field, both the common-path interferometry and the Mach-Zander interferometry were employed in previous works[16–23,25,26]. In these systems, phase stepping was realized by either directly inserting a phase plate[16–18] or exploiting combined superpixels[19–23,25,26]. Given the current bottleneck of the imaging speed is limited by the refresh rate of the illumination rather than the bandwidth of the detector, we propose to implement heterodyne holography to realize phase stepping naturally in time. As shown in Fig. 6, the heterodyne signal oscillates as $\widetilde{a}_n \cos(2\pi\triangle ft + \widetilde{\varphi}_n)$, where a beat frequency $\triangle f$ is introduced between the signal beam and the reference beam. By performing a series of measurement to the beating signal, both the amplitude $a_n$ and the initial phase $\varphi_n$ can be uniquely determined (detailed in the Supplementary Note 1). The major

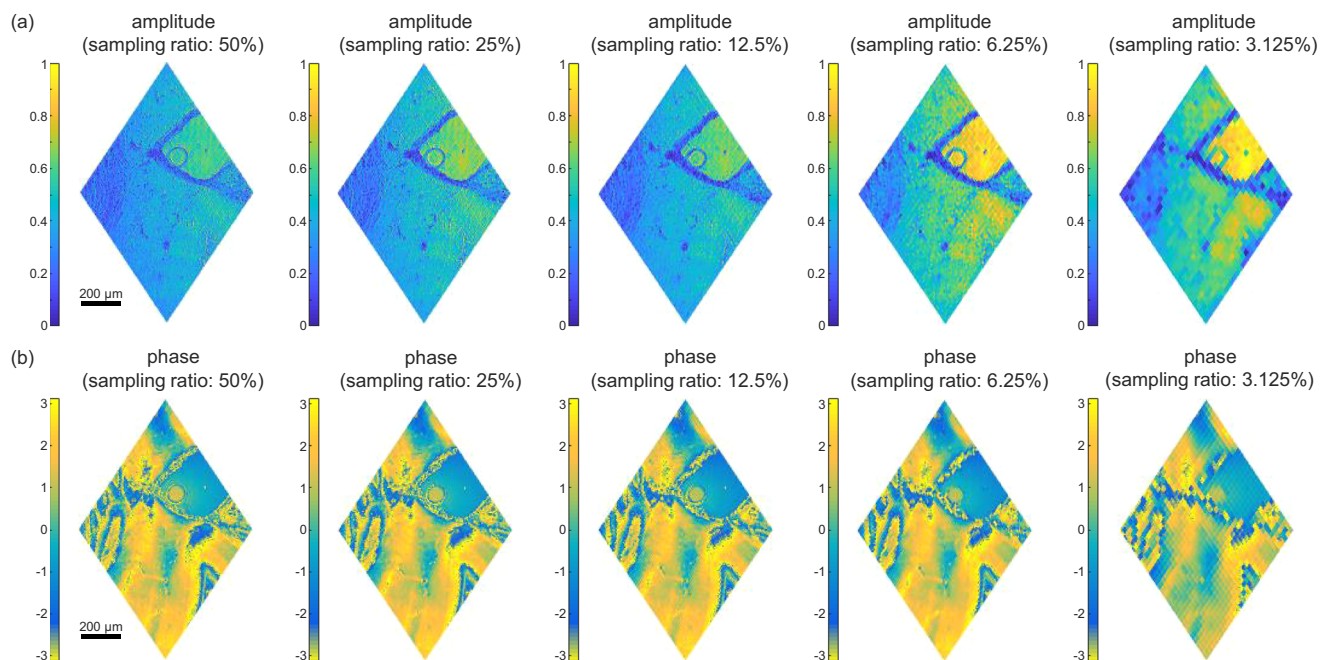

**Fig. 5 Reconstruction of holographic images for the unstained tissue from mouse brains with compressive sensing.** The amplitude and wrapped phase images are reconstructed with different SRs of 50%, 25%, 12.5%, 6.25%, and 3.125%. The corresponding scale bar is 200 μm.

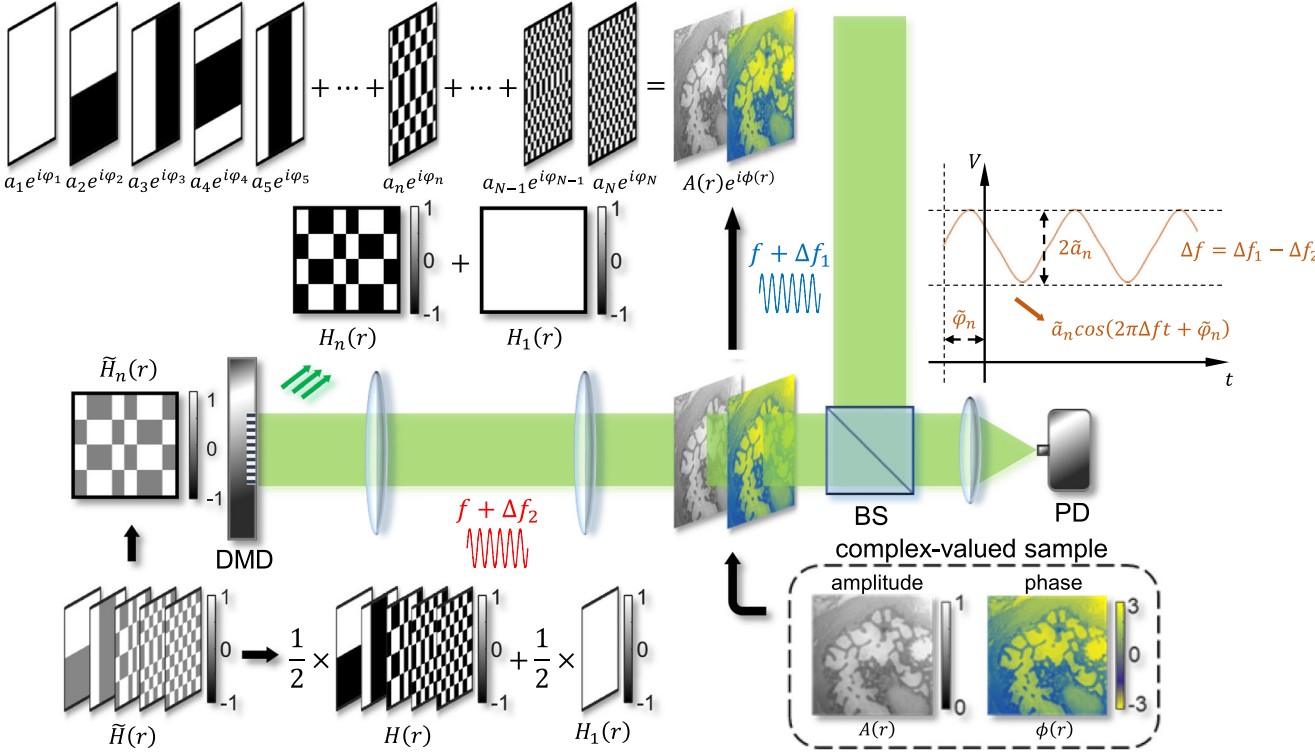

**Fig. 6 Principle diagram of the high-throughput single-pixel compressive holography.** A complex-valued object $O\left(\vec{r}\right) = A\left(\vec{r}\right)e^{i\phi\left(\vec{r}\right)}$ can be expressed as the superposition of a complete set of orthogonal Hadamard basis $H_n\left(\vec{r}\right)$ with corresponding coefficients. To retrieve these coefficients, one can illuminate the object with a series of Hadamard-like patterns $\widetilde{H}_n\left(\vec{r}\right)$ (with components "0" and "1") generated by the DMD. To implement heterodyne holography, a beat frequency $\triangle f$ is introduced between the signal beam and the reference beam, enabling a time-varying signal that can be measured by the photodetector. The simple linear transformation between $\widetilde{H}_n\left(\vec{r}\right)$ and $H_n\left(\vec{r}\right)$ allows the reconstruction of holographic images with pure amplitude patterns. DMD: digital micromirror device; BS: beam splitter; PD: photodetector.

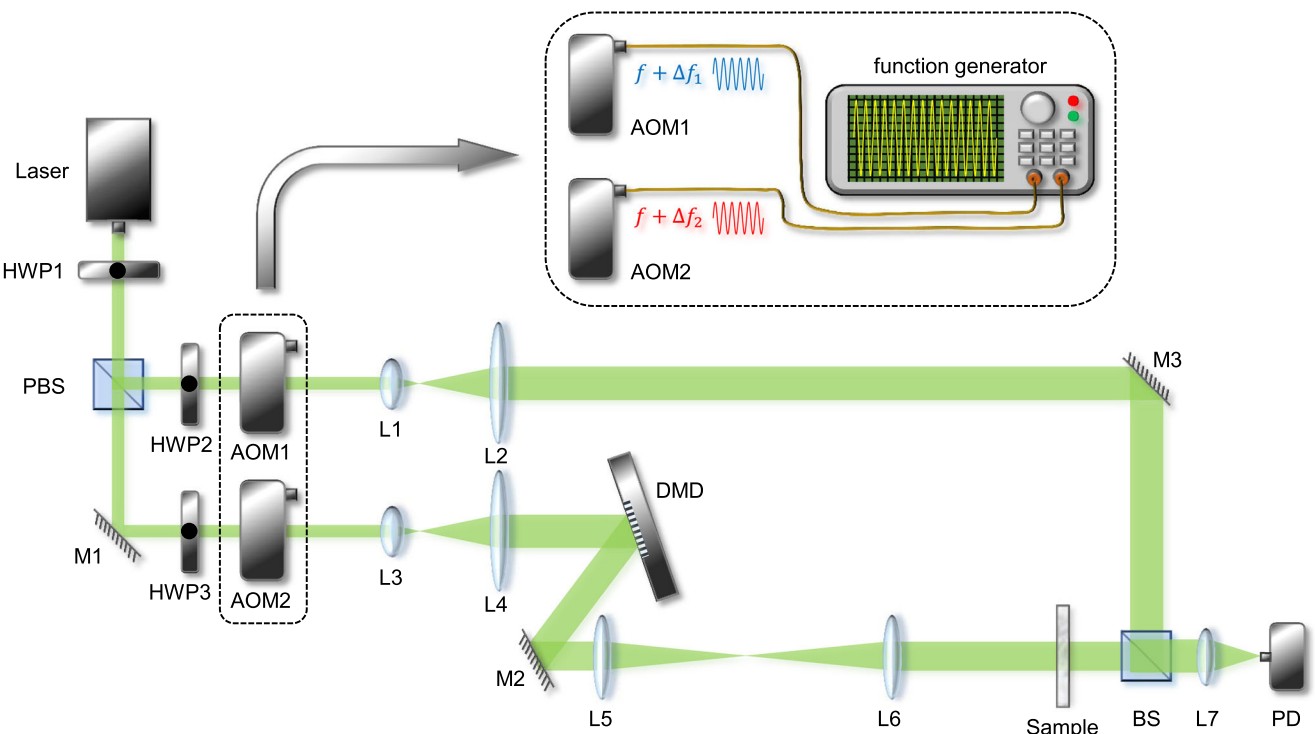

**Fig. 7 Experiment setup of the high-throughput single-pixel compressive holography system.** A series of Hadamard-like patterns were generated and projected to the sample. HWP1-3: half-wave plate; M1-3: mirror; PBS: polarization beam splitter; AOM1-2: acousto-optic modulators that cause a frequency shift to the light passing through; L1-7, lenses ($f_1 = f_3 = 7.5\,mm$, $f_2 = f_4 = 250\,mm$, $f_7 = 150\,mm$, $f_5$ and $f_6$ is scalable to be adapted for various demands); DMD: digital micromirror device that provides amplitude modulation with "0" and "1"; BS: beam splitter; PD: photodetector. The upper inset shows the detailed procedure of how a double-channel function generation generates a beating frequency to drive the AOMs (an electronic power amplifier is omitted here).

advantage of implementing this scheme for holography is that only one Hadamard pattern is displayed for each order as phase stepping is realized naturally in time, thus significantly saving acquisition time and pixel numbers.

**Experimental setup.** The experimental setup is schematically shown in Fig. 7. As a demonstration of principle, a long-coherence solid-state semiconductor laser (MSL-FN-532-100mW, CNI) that operates at 532 nm was used as the light source. After passing through a polarizing beam splitter (PBS), the light source was divided into a signal beam and a reference beam. A half-wave plate was placed in front of the PBS to adjust the intensity ratio of the two beams. To maximize the visibility of the fringes and achieve a good signal-to-noise ratio during experiments, the desired intensity ratio that reaches the single-pixel detector is 50% : 50%. Heterodyne holography was realized by using two AOMs (AOM-505AF1, Intraaction, Optical frequency shift range: $\pm 40 \sim 60$ MHz), which were controlled by a function generator. In this setup, AOM1 in the reference beam shifted optical frequency by 50 MHz + 31,250 Hz, while AOM2 in the signal beam shifted optical frequency by 50 MHz − 31,250 Hz. Thus, the beat frequency of these two beams is 62,500 Hz, indicating a temporal period of 16 μs. After frequency modulation, both beams were expanded by a pair of lenses with 7.5-mm and 250-mm focal lengths. The signal beam was then diffracted by a DMD (V7001 DLP7000&DLPC410) to achieve binary amplitude modulation. This DMD has 768 × 1024 pixels with 13.68-μm pixel size and a 22-kHz refresh rate. Since the DMD diffracts light like a two-dimensional shining grating, the optimum incident and diffracted angles are calculated to be 17.92° and 41.92° (detailed in the Supplementary Note 4). A 4f system that consists of two lenses (scalable due to different operational modes) then imaged the surface of the DMD onto the sample. The sample was placed so that its surface was orthogonal to the optical axis. After interacting with the sample, the signal beam was combined with the reference beam through a beam splitter. It is worth mentioning that since the DMD slightly altered the polarization after modulation, a quarter-wave plate (not shown in the figure) was inserted in the reference beam to maximize the interference visibility. Using a lens with 150-mm focal length, the combined light was collected by a photodetector (DET10A2, Thorlabs) with a bandwidth of 350 MHz, which was then digitized by a data acquisition card (DAC, USB-6251, National Instrument) with a sampling rate of 1.25 Ms/s (not shown in the figure). Considering the 48-μs refresh time of the DMD and the 62,500-Hz beating frequency, three beating cycles last for each Hadamard pattern and 20 data points were acquired within one cycle. We note here that this DMD can support a shorter refresh time down to 45 μs (22 kHz

refresh rate), corresponding to on average 56.8 data points for the duration of one Hadamard pattern. Although such a choice can gain an increase in data rate by roughly 5%, it causes difficulties in batch processing to reconstruct images. What's more, the choice of the beating frequency is also not unique. In practice, there are many choices for beating frequencies such as 125,000 Hz or 250,000 Hz, distributing these 60 data points into 6 and 12 beating cycles, respectively. Notably, for the same number of data points for one displayed Hadamard pattern, the quality of the reconstructed signal should not be sensitive on the choice of the beating frequency, provided the Nyquist sampling criterion was followed. An integer number of beating cycles for each displayed pattern is also desired for computational convenience.

**Summarization of different operational modes.** A step-by-step protocol describing the imaging and reconstruction procedures can be referred to at Protocol Exchange[64]. For different imaging applications, the FOV and the lateral resolution can be adjusted by either choosing the appropriate lens pair in the 4f system or selecting different binning strategies of the pixels. As a concrete example, the two lenses in the 4f system shown in Fig. 7 have the same focal length of 125 mm, indicating that Hadamard patterns displayed by the DMD were 1:1 imaged onto the surface of the sample. In this condition, if we make full use of the active area by adopting the strategy of binning 3 × 3 pixels into 256 × 256 super-pixels, this system is expected to have a large FOV of 14.9 mm × 11.1 mm with a lateral resolution of 58.0 μm × 43.1 μm (large-FOV mode). The difference in the FOV and lateral resolution along different directions is caused by the 45°-tilted arrangement of the DMD, which is described in detail in the Supplementary Note 4. The operation of the imaging system can also be altered into a high-resolution mode for microscopy by employing two lenses with 300-mm and 30-mm focal lengths in the 4f system. Since Hadamard patterns were minified by 10 times, the lateral resolution becomes 5.80 μm × 4.31 μm. Nonetheless, since we fixed the highest order of Hadamard patterns to be 256 × 256 during experiments (limited by the available RAM of the DMD), the throughput of the holographic system remains the same for different modes.

As a final remark, we briefly summarize the performance of several operational modes developed in this work, which are listed in Table 2. Two large-FOV modes and one high-resolution mode were demonstrated with the same illumination device, beat frequency, and the sampling rate of DAC. Thus, the same SBP-T was achieved for all three operational modes, calculated by multiplying the reciprocal of refresh time of the illumination device by 2. Moreover, the two large-FOV modes

**Table. 2 List of the parameters in both the large-FOV mode and the high-resolution mode.**

|  | Large-FOV mode 1 (Main Text) | Large-FOV mode 2 (Supplementary Information) | High-resolution mode 1 (Main Text & Supplementary Information) |
|---|---|---|---|
| Pixel size (µm) | 13.68 |  |  |
| Refresh time (ms) | 0.048 |  |  |
| Beating frequency (Hz) | 62,500 |  |  |
| Sampling rate of DAC (Ms/s) | 1.25 |  |  |
| Strategy of binning pixels | 768 × 768 pixels with 3 × 3 binning | 512 × 512 pixels with 2 × 2 binning | 768 × 768 pixels with 3 × 3 binning |
| Front lens in 4f system | AC254-125-A, Thorlabs (125 mm) | AC254-125-A, Thorlabs (125 mm) | AC254-300-A, Thorlabs (300 mm) |
| Rear lens in 4f system | AC254-125-A, Thorlabs (125 mm) | AC254-125-A, Thorlabs (125 mm) | AC254-030-A, Thorlabs (30 mm) |
| Lateral resolution (µm) | 58.0 × 43.1 | 38.7 × 28.8 | 5.80 × 4.31 |
| FOV (mm) | 14.9 × 11.1 | 9.91 × 7.37 | 1.49 × 1.11 |
| SBP-T (pixels/s) | 41,666.6 |  |  |

share the same 4f system, while the only difference is the binning strategy. Due to this reason, the lateral resolution and the FOV of these two modes are simply scaled by a factor of 1.5. As for the high-resolution mode, a different 4f system was used. Therefore, the lateral resolution and the FOV of this mode are scaled to be 10 times smaller than that in the first large-FOV mode. Detailed evaluations on these parameters can be found in the Supplementary Note 11. Currently, the switching between the large-FOV mode and the high-resolution mode requires hardware modification of the 4f optical setup, which is inconvenient and prevents real-time operation. Nonetheless, we anticipate that a variable 4f setup that consists of tunable lenses and an advanced electromechanics system with auto-focusing can enable smooth adjustment between different operational modes in the future.

**Statistics and reproducibility**. To ensure the authenticity and reliability of the holographic imaging performances, those imaging experiments mentioned in Main Text and Supplementary Information are repeated at least 4 times, showing similar results. Besides, micrographs obtained with the bright field microscope are captured once or twice. As this commercialized equipment (Axio Scope, ZEISS) produces almost identical results, these micrographs are regarded as golden standards for comparing.

**Preparation of biological samples**. Two 3-month-old female mice (C57BL/6N-Hsd: Athymic Nude-FoxlNU, Harlan) were sacrificed with an overdose of pentobarbital (100 mg kg$^{-1}$). Mouse tails and brains were collected, immersed in 4% paraformaldehyde solution for 24 h at 4 °C, and decalcified in ethylenediaminetetraacetic acid decalcifying solution for four weeks. Then, they were embedded into paraffin blocks, sectioned into slices with different thicknesses. Dyeing operation was applied only to the slices of mouse tails using hematoxylin and eosin. Finally, those biological tissues, including both stained mouse tails and unstained mouse brains, were collected and fixed onto glass slides. We affirmed that all procedures were carried out in conformity with ethical regulations approved by the Institutional Animal Care and Use Committee, Sun Yat-sen University.

**Reporting summary**. Further information on research design is available in the Nature Research Reporting Summary linked to this article.

# Data availability
Data to produce holographic images in this work have been deposited in public repository Zenodo, which are provided with this paper[60].

# Code availability
Code used for imaging reconstruction in this work can be accessed in public repository Zenodo[65].

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

## Acknowledgements

Y.S. thank Dr. Xiaoming Wei at South China University of Technology for fruitful discussions on imaging techniques. This work was supported in part by National Key Research and Development Program of China (2019YFA0706301), National Natural Science Foundation of China (12004446, U2001601), and Open Fund of State Key Laboratory of Information Photonics and Optical Communications (Beijing University of Posts and Telecommunications) (IPOC2020A003).

## Author contributions

Y.S. conceived the idea and initiated the project. Y.S. and D.W. designed the experiments. D.W. built the optical system and performed the experiments. D.W., J.L., and Y.F. designed the system control. D.W., G.H., and X.F. performed the simulation and processed both the simulation and experimental data. R.Z. prepared biological samples. Y.S. and Z.L. provided overall supervision.

## Competing interests

The authors declare no competing interests.
