## [Peer Review File · Nature Communications]

Reviewers' Comments:

Reviewer #1:

Remarks to the Author:

The authors present an optical heterodyne detection scheme for single-pixel holographic imaging. Single-pixel holography is not new (see for example references 13-20 in the manuscript) but the role of heterodyne detection has not been studied in depth previously.

As a matter of fact, frequency beating between the two arms of the Mach-Zehnder interferometer enables phase sensing using the amplitude binary codification natural in digital micromirror devices. Neither phase-shifting (which enlarges signal acquisition), nor computer-generated hologram codification (which sacrifices spatial resolution) are compromised. Based on the above architecture, there is a claim for an improvement in the space-bandwidth-time product.

However, the above trade-off has been explored several times in the past for single-pixel imaging, both at the hardware and the software level. In Sci Rep. 8:2369 (2018) video operation at 30 frames/sec for a 128x128 image at a frame rate of 30 frames/sec is demonstrated. Extrapolation to single-pixel holography is straightforward.

Switching between large field-of-view and high-resolution mode is also claimed. However this is done at the expense of a hardware modification of the 4f optical setup which prevents real-time operation.

As the authors mention, the use of compressive sensing (the possibility of compressing during the measurement process) is an outstanding feature of single-pixel holography. The theory of compressive sensing relies on two principles: the sparsity of the signal of interest, and the incoherence between the bases employed for the measurement and the reconstruction. It should be convenient that the authors clarify if the "incoherence" condition is essential for making compressive sensing work.

Concerning, parameters in Table 1 about state-of-the art in single-pixel imaging and single-pixel holography I miss some important papers such as Opt. Express 28, 28190-28208 (2020) and Nat. Photon. 13, 13-20 (2019). Also, I find convenient to include some references for the single-pixel imaging () or for non-interferometric single-pixel phase imaging (Optica 5, 164-174 (2018)). Finally, adaptive and smart sensing should be considered for a fair comparison between the spatial and temporal trade-off in single-pixel imaging.

In summary, the approach and the theoretical description are good. However, all of the results can easily be derived from the combination of two well-established techniques: single-pixel holography and heterodyne holography. Also, the quality of experimental implementation should be improved to support the claims in the abstract concerning biological applications.

In this sense, everything is highly predictable and the paper lacks of the novelty required for publication in Nature Communications.

Reviewer #2:

Remarks to the Author:

In this paper, Wu et al. report single-pixel compressive holography for multi-scale imaging. Developed upon heterodyne holography and Hadamard encoding, the reported SPH system achieves a space-bandwidth-time product (SBP-T) of 41,667 pixels/s. Compared with existing prototypes, the reported system delivers superior specifications in throughput, pixel counts, and field of view (in the large FOV mode), as well as has a comparable micrometer-level spatial resolution (in the high-resolution mode). This technique is applied to image a cross-section of a mouse tail. The capability of resolving both amplitude and phase allows this technique to reveal various features on the sample.

This work can be published in Nature Communications. The main breakthrough in this work is the implementation of heterodyne detection to nullify the necessity of phase-shifting measurement.

This is a clever trick to overcome the speed limitation imposed by the DMD. The technical specifications of this system are among the best in the field. The authors have also expanded the application of this technique to bio-imaging. The technical innovations included in this work can benefit the relevant communities.

I have the following comments for the authors to address.

1. The literature review of single-pixel imaging could be strengthened. As the authors wrote in the introduction, single-pixel cameras have great performance across almost the entire spectrum range. Some previous papers for applying single-pixel imaging in infrared (e.g., *Opt. Express* 25, 2998-3005 (2017)), THz (e.g., *Nat. Commun.* 11, 2535 (2020)), and even photoacoustic imaging (e.g., *Opt. Lett.* 39, 430-433 (2014)) could be included. Besides, a recent trend in single-pixel cameras is 3D imaging. Representative works in this category (e.g., *Nat. Commun.* 7, 12010 (2016), *Opt. Express* 28, 29377-29389 (2020), and *APL Photonics* 5, 020801 (2020)) could be included.

2. In line 15-17 on Page 3: "For example, we can operate under large-FOV mode (14.9 mm × 11.1 mm) to monitor the environment [9, 10, 21] or switch to high-resolution mode (5.8 μm × 4.3 μm) to scrutinize microstructures [18, 20]." What determines the limit of the FOV and resolution in this work? Have the authors pushed these numbers to the limit already? Could the authors provide some justifications on how to get these numbers? Also, how would the spatial resolution in amplitude and phase be degraded with the compressed ratio?

3. The authors claimed that the strategy of 768 × 768 pixels with 3 × 3 binning from DMD was performed in holographic imaging. Why not make full use of the entire DMD face with 768 × 1024 pixels and 3 × 4 binning? This strategy should provide a better SNR and imaging quality.

4. In Fig. 2, lens L7 is regarded as a Fourier transformation of the input field. When loading patterns with high spatial frequency, the output field is no longer a single point but exhibits spatial distribution. Considering the finite sense area of the single-pixel detector, it is possible that not all the output energy is collected. Are there any considerations or calculations to consider this condition?

5. In the table on Page 3, the column of "refresh time" listed in this work is 48 μs, indicating that the full speed of the DMD (22 kHz) was not fully used. Why not making full use of the 22 kHz refresh rate to push the SBP-T to the limit?

6. In the provided files "data_and_supporting_files.zip" and "readme.pdf", it is mentioned that "To minimize phase drifting caused by environmental disturbance during the experiment.....", could the authors provide a detailed explanation on the reasons for inserting additional Hadamard patterns and how to process these data corresponding to additional patterns?

7. In Line 17 on Page 4, Eq. (3), the first order of Hadamard-like patterns is simply a DC signal. My understanding is that if this value was not subtracted for each coefficient as the authors did, the final image would have a constant background. Nonetheless, for imaging purposes, a constant background is acceptable.

8. What's the bandwidth of the photodiode? Specifications of this component are helpful for readers to adopt this technique.

9. In Fig. 2, the DMD is not parallel to the sample plane. Will this introduce any measurement error or distortion?

10. The gold standard is missing to cross-check the presented technique. Can the authors use existing techniques (e.g., phase-contrast microscopy) to verify their results? Also, what's the measurement accuracy? This would be another important parameter that would determine the application scope of this technique.

11. In Fig. 5, the authors used red dashed circles to point out show some supplement information

in the phase image from the amplitude counterpart. This part could be elaborated on because it showcases the capability of this technique.

12. A few typos are found in the manuscript. For example, by "mussel", I believe the authors intended to say "muscle".

Reviewer #3:

Remarks to the Author:

Dear Authors,

In this manuscript, a novel technique for single pixel compressive holography is proposed. The proposed technique employs heterodyne interference to achieve high-throughput measurement.

The authors construct an optical system based on the proposed technique. And experimental verifications are described.

In the experimental verifications, parts of a USAF chart for estimation of spatial resolution and biological tissues are reconstructed to demonstrate the usefulness of the system.

I have some questions and comments to improve the manuscript.

(1) In Fig.2 : Why the experimental set up consists of two AOMs?

I think that heterodyne interferometry can be implemented with only one AOM with easy control. Also, the authors should describe the reason why the beat frequency is set to 62,500 Hz.

(2) About Fig. 4 (a): the authors should discuss factors that the image quality at sampling ratio 50% is worse than that in more critical case (25% & 12.5%).

The discussions are important for judgement which the used algorithm for image reconstruction is. Such Discussion are important in determining whether the algorithm used for image reconstruction is practical.

(3) About holographic imaging:

To show the effectiveness as holographic imaging, phase objects or three-dimensional objects should be experimentally verified.

This is because verification for phase imaging with the constructed experimental setup is not enough.

(On the other hand, That for amplitude reconstruction is quite enough to show the usefulness.)

(4) In conclusion: some performances of the proposed method are shown.

Those estimations are much important in the paper.

The authors should summarize the performance as a table like Table 1 and discuss on relations between scalability and specifications of the elemental devices in detail.

(5) Forms of reconstructed images in Figs. 3, 4, 5, and 6 : (This comment may be not so essential.)

Why area of measurement is not square like but diamond shape (rhomboid) ?

We thank the editor for organizing the review and all three reviewers for their valuable comments. We have carefully digested the reviewers' comments and largely improved both main text and Supplementary Information in the revised submission.

The major changes to the manuscript can be summarized as the following points:

1. We improved the biological application by performing new experiments to image unstained tissue from mouse brains and supplementing corresponding holographic images in the revised submission. Slices of unstained mouse brains with various thicknesses, including 10 μm , 80 μm , 100 μm , and 120 μm , were imaged. These unstained slices of biological tissue show relatively poor contrast in amplitude but exhibit rich information with good contrast in phase. These new experimental results demonstrate the capability of the high-throughput single-pixel holography to image biological tissue that contains rich information in phase.
2. We improved the biological application by performing additional experiments to image another slice of stained tissue from mouse tails. Corresponding holographic images are provided in Supplementary Information of the revised submission.
3. We added a new experiment to quantify the performance of the holographic system in phase by imaging a quantitative phase resolution target. Using the obtained holographic images, we quantified the spatial resolution in phase, confirming its agreement with the theoretical resolution of the system. We also compared the reconstructed phase value to the actual phase value of the resolution target. The phase error was quantified to be about 0.104 rads ($\leq \lambda/60$). This new experimental result demonstrates the high-throughput single-pixel holography can quantitatively image phase objects.
4. We added a Table to summarize and compare the parameters of different operational modes, i.e., the large field-of-view (FOV) mode and the high-resolution mode, employed in this work.
5. We performed numerical simulations to study the sampling ratio of compressive sensing under the influence of measurement noises. Both numerical and experimental results indicate that the SR of 25% is likely to be the suitable choice under the current experimental condition.
6. We added detailed descriptions to explain the choice of beat frequency and the operational condition of the digital micromirror device.
7. We added a paragraph to describe the preparation procedures of these biological samples. We added Mr. Runsen Zhang as a coauthor, who assisted in the preparation of these biological samples. Additional fundings that supported the experiments during the revision process were added in the acknowledgment.
8. Since new experimental results to image biological samples were added, we slightly modified corresponding descriptions in the abstract and discussion section. We also added the sections of Data availability, Code availability, Acknowledgments, Author contributions, Competing interests to fulfill the requirement of the journal.

A point-by-point response to the reviewers' comments is provided in the following.

Point-by-point responses to reviewers' comments

Reviewer #1:

The authors present an optical heterodyne detection scheme for single-pixel holographic imaging. Single-pixel holography is not new (see for example references 13-20 in the manuscript) but the role of heterodyne detection has not been studied in depth previously.

1. As a matter of fact, frequency beating between the two arms of the Mach-Zehnder interferometer enables phase sensing using the amplitude binary codification natural in digital micromirror devices. Neither phase-shifting (which enlarges signal acquisition), nor computer-generated hologram codification (which sacrifices spatial resolution) are compromised. Based on the above architecture, there is a claim for an improvement in the space-bandwidth-time product. However, the above trade-off has been explored several times in the past for single-pixel imaging, both at the hardware and the software level. In Sci Rep. 8:2369 (2018) video operation at 30 frames/sec for a 128x128 image at a frame rate of 30 frames/sec is demonstrated. Extrapolation to single-pixel holography is straightforward.

Response: We thank the reviewer for this comment and for suggesting this important reference. This reference reports real-time single-pixel imaging that provides intensity-based images. The claimed video frame rate (30 frames-per-second) with 128×128 pixels is enabled by compressive sensing with a compression ratio of only 2%. A deep convolutional auto-encoder network (DCAN) was designed to optimize the sequence of the Hadamard basis and reconstruction model. Thus, the main achievement of this reference is to facilitate the imaging reconstruction process, which is completely different from the main purpose of our work (high-throughput). Moreover, for imaging acquisition, the throughput, i.e., the space-bandwidth-time product, of the system developed in this reference is solely determined by the parameters of the digital micromirror device, which is the same as those reported in previous works listed in Table 1 of the revised main text. Nonetheless, the algorithm developed in the suggested reference can be borrowed into our work to facilitate imaging reconstruction. Given enough training data with holographic images of biological samples in the future, we envision that this deep learning assisted reconstruction algorithm can effectively shorten the time cost for imaging reconstruction of our developed single-pixel holography. Thus, we added this reference and discussed this possibility of facilitating reconstruction speed using deep learning in the Discussion section of the revised main text: **Due to the huge computational burden of imaging reconstruction, real-time imaging of live cells with time-varying features is still unachievable with the current holographic system. We envision that this constraint can be potentially alleviated through the assistance of deep learning and compressive sensing [62].**

2. Switching between large field-of-view and high-resolution mode is also claimed. However, this is done at the expense of a hardware modification of the 4f optical setup which prevents real-time operation.

Response: We thank the reviewer for this comment and agree that the current holographic system requires modifying the 4f optical setup to switch between the large field-of-view mode and the high-resolution mode. In this work, we did not claim real-time operations that can switch between different operational modes. This kind of hardware modification is not unique to our system but is commonly seen in many microscopes as well. In addition, this hardware limitation can be conquered by using tunable lenses and an advanced electromechanical system with auto-focusing in future works. To address this issue, we added the following descriptions in the Method section of the revised main text to clarify this point: **Currently, the switching between the large FOV mode and the high-resolution mode requires hardware modification of the 4f optical setup, which is inconvenient and prevents real-time operation. Nonetheless, we anticipate that a variable 4f setup that consists of tunable lenses and an advanced electromechanics system with auto-focusing can enable smooth adjustment between different operational modes in the future.**

3. As the authors mention, the use of compressive sensing (the possibility of compressing during the measurement process) is an outstanding feature of single-pixel holography. The theory of compressive sensing relies on two principles: the sparsity of the signal of interest, and the incoherence between the bases employed for the measurement and the reconstruction. It should be convenient that the

authors clarify if the “incoherence” condition is essential for making compressive sensing work.

Response: We thank the reviewer for this valuable suggestion and we take the liberty to assume the incoherence is defined using formula 1.7 in the paper “*Sparsity and incoherence in compressive sampling, Inverse Problems 23, 969, 2007*”, written by Emmanuel Candès and Justin Romberg. Here, incoherence is the estimation for correlation between the bases for measurement (sensing modality) and reconstruction (signal model). There are two major approaches to realize compressive sensing. The first one is to employ an iterative converging algorithm with different kinds of regularization factors, such as the L1-norm and total variation (TV) factor. For this case, the employed bases for measurement and reconstruction can be either “coherence” or “incoherence”. When no prior information of the sample was provided, a random ordering was preferred, and “coherence” should generally provide good reconstruction results. Nonetheless, when an optimized ordering for a sparse signal is known prior, “incoherence” could lead to better reconstruction results with a small number of measurements. The second one is to use the same set of bases for measurement and reconstruction. An inverse transformation like inverse Hadamard transformation or inverse Fourier transformation is employed to reconstruct the image. By definition, compressive sensing that follows this rule can be categorized as “incoherence”. Our work belongs to the second approach of compressive sensing, which employs the same set of Hadamard bases for both measurement and reconstruction. A given ordering strategy of the square path was adopted to simplify the reconstruction process. Therefore, the “incoherence” condition is essential to make compressive sensing work in our case. To clarify this point, we added the following sentence in the Results section of the revised main text: “**In this work, an ordering of square sampling path (detailed in the Method section) was applied for sampling, and a direct inverse fast Hadamard transformation using the same set of bases was adopted for reconstruction. Thus, incoherence is confirmed between the bases employed for the measurement and the reconstruction, which is essential to make compressive sensing work for our holographic system [61].**”

4. Concerning, parameters in Table 1 about state-of-the art in single-pixel imaging and single-pixel holography I miss some important papers such as Opt. Express 28, 28190-28208 (2020) and Nat. Photon. 13, 13-20 (2019). Also, I find convenient to include some references for the single-pixel imaging () or for non-interferometric single-pixel phase imaging (Optica 5, 164-174 (2018)). Finally, adaptive and smart sensing should be considered for a fair comparison between the spatial and temporal trade-off in single-pixel imaging.

Response: We thank the reviewer for supplementing these important references. We studied the two suggested review articles on this topic (Opt. Express 28, 28190-28208 (2020) and Nat. Photon. 13, 13-20 (2019)). Both references were added in the revised main text as important references to prepare Tab. 1: **Based on the review articles covering this topic [54, 55], Tab. 1 summarizes the performance of some representative works about SPI [8, 12-15, 24] and SPH [16-23, 25, 26] reported in the literature.**

The non-interferometric single-pixel phase imaging (Optica 5, 164-174 (2018)) was also added as an important reference in the Introduction section of the revised main text: **Borrowing the concept of the Shack-Hartmann sensor, a non-interferometric phase image was demonstrated using a single-pixel detector [32]. However, the special requirement of using a lateral position detector sacrifices the advantages that conventional SPH holds.**

As suggested by the reviewer, we also added a description to discuss adaptive and smart sensing for the spatial and temporal trade-off in single-pixel imaging in the Introduction section of the revised main text: **Recently, adaptive and smart sensing with dynamic supersampling was reported to combine with compressive sensing in SPI. The enabling feature of this approach is to rapidly record fast-changing features by dynamically adapting to the evolution of the scene. Thus, it significantly shortens acquisition time without considerably sacrificing spatial information [31]. Since the performance of adaptive and smart sensing largely depend on the sparsity and types of samples, Tab. 1 only compares representative works on SPI and SPH from the perspective of the system (or hardware) rather than the algorithm. This point was clarified in the Introduction section of the revised main text: “We note here that compressive sensing, including adaptive and smart sensing, is effective in breaking the spatial and temporal trade-off for most SPI and SPH systems. However, the improvement in SBP- T is ambiguous to be quantified, especially when the target sample is not specified. Therefore, compressive sensing was not considered when estimating the parameters displayed in Tab. 1 for a fair comparison.”**

5. In summary, the approach and the theoretical description are good. However, all of the results can easily be derived from the combination of two well-established techniques: single-pixel holography and heterodyne holography. Also, the quality of experimental implementation should be improved to support the claims in the abstract concerning biological applications. In this sense, everything is highly predictable and the paper lacks of the novelty required for publication in Nature Communications.

Response: We thank the reviewer for the recognition of the approach and the theoretical description of this work. We largely improved the claimed biological applications in the revised submission, by adding new experiments to imaging slices of both stained mouse tails and unstained mouse brains with various thicknesses. High-resolution holographic imaging results of these biological samples reveal rich information in both amplitude and phase, showing the prospect of our work. We believe that these newly supplement holographic results can support the claim of biological applications.

Detailed descriptions on imaging unstained biological tissue were added as a new paragraph “**Holographic imaging of unstained tissue from mouse brains**” in the Results section of revised main text and a section of “**Additional holographic images of unstained tissue from mouse brains**” in revised Supplementary Information.

In revised main text

Holographic imaging of unstained tissue from mouse brains. Having demonstrated holographic imaging of stained tissue from mouse tails, we then switched to image unstained tissue from mouse brains. Generally, images of unstained tissue exhibit low contrast in amplitude, but providing sufficient contrast through phase imaging. Before proceeding, we quantified the performance of the holographic system in phase by imaging a quantitative phase resolution target (QPT, BenchMark Tec). Figure 5(a) shows the reconstructed phase image of the resolution target, containing groups 6 and 7. To quantify the resolving power in phase, the corresponding 1D profile denoted by the black bracket within the resolution target, crossing element 6 of group 6 (4.386- μm width), is shown in the upper inset. Three verticle bars can be differentiated, indicating that the theoretical resolution under the high-resolution mode (5.80 $\mu\text{m} \times 4.31 \mu\text{m}$) was achieved in phase. We further quantified the accuracy of the measurement in phase by estimating the phase difference between the bar enclosed by a dashed box at the bottom of the target and its adjacent background. 1D profile that crosses both the bar and the background is shown in the lower inset using blue circles, exhibiting a stepped structure. The averaged phase difference was estimated to be $\Delta\varphi_{\text{exp}} \approx 1.691$ rads, yielding a phase error of only 0.104 rads ($\leq \lambda/60$) compared to the actual phase difference (1.795 rads). Detailed analysis on imaging the quantitative phase resolution target can be found in section VII of Supplementary Information. Then, we proceed to image unstained biological tissue. Figure 5(b) shows the image of an 80- μm -thick slice of unstained tissue from mouse brains, captured using a bright-field microscope. In this image, some representative types of tissue were denoted with black arrows, such as piriform cortex, perirhinal cortex, olivary pretectal nucleus, and white matter. For unstained tissue, it is usually challenging to identify different structures in amplitude, due to the insufficient contrast in transmission. Figures 5(c), (d), and (e) show a series of reconstructed holographic images for different parts of the mouse brain, indicated by three labeled diamond-shaped boxes in Fig. 5(b). Same as the one captured using the conventional microscope, these amplitude images only manifest rough outlines with low contrast. By comparison, phase images show rich details that can be hardly seen in their amplitude counterparts. For example, the piriform cortex and the perirhinal cortex are identifiable in phase images. Data used for producing Figs. 5(c)-(e) are provided in Supplementary Data 2. These results demonstrate the capability of employing SPH to image relatively transparent biological tissue.

Fig. 5 Performance of the high-throughput SPH with unstained tissue from mouse brains in high-resolution mode. (a) Reconstructed phase image of a quantitative phase target. The upper inset: the corresponding one-dimensional (1D) profile of element 6 of group 6; the lower inset: quantitative analysis on the accuracy of the reconstructed phase values, resulting in a phase error of 0.104 rads. (b) The image of a slice of unstained tissue from mouse brains, captured using a conventional microscope. Three diamond-shaped boxes represent the area being imaged by the holographic system. (c)-(e) The reconstructed amplitude and phase images for different parts of the unstained tissue. The corresponding scale bar is 200 μm .

For the selected area in Fig. 5(c), a series of holographic images reconstructed with compressive sensing at various SRs are illustrated in Fig. 6 as well. Similar to before, these images show the effectiveness of compressive sensing in dealing with biological tissue that contains rich information in phase. We also applied SPH to image other pieces of unstained tissues from mouse brains with different thicknesses, ranging from 10 μm to 120 μm , with details shown in section IX of Supplementary Information.

Fig. 6 Reconstruction of holographic images for the unstained tissue from mouse brains with compressive sensing. The amplitude and wrapped phase images are reconstructed with different sampling ratios of 50%, 25%, 12.5%, 6.25%, and 3.125%. The corresponding scale bar is 200 μm .

In revised Supplementary Information

Additional holographic images of unstained tissue from mouse brains

To show the imaging capability for unstained tissue, we also imaged a 100- μm -thick slice of unstained mouse brain. Figure 9(a) shows a bright-field image of this slice, captured using a conventional microscope. The imaged part of the tissue contains white matter and grey matter. Since the unstained slice is relatively thick with roughly the same transmission across the imaging region, it is challenging for the conventional microscope to provide good contrast. Figures 9(b), (c), and (d) show a series of reconstructed holographic images for different parts of the mouse brain, denoted by three labeled diamond-shaped boxes in Fig. 9(a). As expected, the features in amplitude images match well with that at corresponding areas in Fig. 9(a), which do not provide too much information with good contrast. As a comparison, phase images provide much better contrast, revealing many detailed structures that are indiscernible in their amplitude counterparts.

Fig. 9 Holographic performance of unstained mouse brain tissue with 100- μm thickness. (a) The image of a slice of 100- μm -thick unstained tissue from mouse brains, captured using a conventional microscope. Three diamond-shaped boxes represent the area being imaged by the holographic system. (b)-(d) The reconstructed amplitude and phase images for different parts of the unstained tissue. The corresponding scale bar is 200 μm .

We also imaged an even thicker slice of unstained mouse brain with a thickness of 120 μm . Figure 10(a) shows a bright-field image of this slice, captured using a conventional microscope. The imaged part contains the ventral part of the lateral septal nucleus, inferior colliculus, and white matter. Figures 10(b), (c), and (d) show a series of reconstructed holographic images for different parts of mouse brain, denoted by three labeled diamond-shaped boxes in Fig. 10(a). Again, phase images provide much better contrast, compared to their amplitude counterparts.

Fig. 10 Holographic performance of unstained mouse brain tissue with 120- μm thickness. (a) The image of a slice of 120- μm -thick unstained tissue from mouse brains, captured using a conventional microscope. Three diamond-shaped boxes represent the area being imaged by the holographic system. (b)-(d) The reconstructed amplitude and phase images for different parts of the stained tissue. The corresponding scale bar is 200 μm .

Very thin unstained tissue from mouse brains down to 10- μm thick was also imaged using our holographic system. Figure 11(a) shows a bright-field image of this slice, captured using a conventional microscope. The imaged part contains white matter and grey matter. Figures 11(b), (c), and (d) show a series of reconstructed holographic images for different parts of mouse brain, denoted by three labeled diamond-shaped boxes in Fig. 11(a). For such a thin tissue, it is hard to see details through the amplitude images. Nonetheless, phase images still provide rich information with good contrast.

Fig. 11 Holographic performance of unstained mouse brain tissue with 10- μm thickness. (a) The image of a slice of 10- μm -thick unstained tissue from mouse brains, captured using a conventional microscope. Three diamond-shaped boxes represent the area being imaged by the holographic system. (b)-(d) The reconstructed amplitude and phase images for different parts of the stained tissue. The corresponding scale bar is 200 μm .

Detailed descriptions on imaging additional slices of stained biological tissue from mouse tails were added as a new section “**Additional holographic images of stained tissue from mouse tails**” in revised Supplementary Information.

Additional holographic images of stained tissue from mouse tails

To further demonstrate the imaging capability for stained biological tissue, we imaged another piece of slice from mouse tails,

which is 10- μm thick. Figure 8 shows a bright-field image of this slice, captured using a conventional microscope. In this image, several types of tissue such as muscle, cortical bone, and cancellous bone are visually identified. Figures 8(b), (c), and (d) show a series of reconstructed holographic images for different parts of mouse tail. Again, for this stained slice, the amplitude images are in good agreement with the one shown in Fig. 8(a), manifesting great distinctions among different types of tissue. Like the one presented in Fig. 3 of main text, the reconstructed phase images are analogous to their amplitude counterparts.

Fig. 8 Holographic performance of stained mouse tail tissue with 10- μm thickness. (a) The image of a slice of a 10- μm -thick stained tissue from mouse tails, captured using a conventional microscope. Three diamond-shaped boxes represent the area being imaged by the holographic system. (b)-(d) The reconstructed amplitude and phase images for different parts of the stained tissue. The corresponding scale bar is 200 μm .

Reviewer #2:

In this paper, Wu et al. report single-pixel compressive holography for multi-scale imaging. Developed upon heterodyne holography and Hadamard encoding, the reported SPH system achieves a space-bandwidth-time product (SBP-T) of 41,667 pixels/s. Compared with existing prototypes, the reported system delivers superior specifications in throughput, pixel counts, and field of view (in the large FOV mode), as well as has a comparable micrometer-level spatial resolution (in the high-resolution mode). This technique is applied to image a cross-section of a mouse tail. The capability of resolving both amplitude and phase allows this technique to reveal various features on the sample. This work can be published in Nature Communications. The main breakthrough in this work is the implementation of heterodyne detection to nullify the necessity of phase-shifting measurement. This is a clever trick to overcome the speed limitation imposed by the DMD. The technical specifications of this system are among the best in the field. The authors have also expanded the application of this technique to bio-imaging. The technical innovations included in this work can benefit the relevant communities.

I have the following comments for the authors to address.

1. The literature review of single-pixel imaging could be strengthened. As the authors wrote in the introduction, single-pixel cameras have great performance across almost the entire spectrum range. Some previous papers for applying single-pixel imaging in infrared (e.g., Opt. Express 25, 2998-3005 (2017)), THz (e.g., Nat. Commun. 11, 2535 (2020)), and even photoacoustic imaging (e.g., Opt. Lett. 39, 430-433 (2014)) could be included. Besides, a recent trend in single-pixel cameras is 3D imaging. Representative works in this category (e.g., Nat. Commun. 7, 12010 (2016), Opt. Express 28, 29377-29389 (2020), and APL Photonics 5, 020801 (2020)) could be included.

Response: We thank the reviewer for providing these important references. We included them in the Introduction section of the revised main text.

a. Opt. Express 25, 2998-3005 (2017), Nat. Commun. 11, 2535 (2020), and Opt. Lett. 39, 430-433 (2014)

Enabled by this property, SPI has been demonstrated with great success when operating with infrared light [1], Terahertz wave [2], and even photoacoustic signal [3].

b. Nat. Commun. 7, 12010 (2016), Opt. Express 28, 29377-29389 (2020), and APL Photonics 5, 020801 (2020)

By employing various coding mechanisms including Hadamard bases [8, 11-23], Fourier bases [12, 24-26], and random patterns [27], SPI has also been extended and demonstrated with applications in full-color imaging [13], multispectral imaging [14], time-resolved imaging [15], and three-dimensional imaging [28-30].

2. In line 15-17 on Page 3: “For example, we can operate under large-FOV mode (14.9 mm × 11.1 mm) to monitor the environment [9, 10, 21] or switch to high-resolution mode (5.8 μm × 4.3 μm) to scrutinize microstructures [18, 20].” What determines the limit of the FOV and resolution in this work? Have the authors pushed these numbers to the limit already? Could the authors provide some justifications on how to get these numbers? Also, how would the spatial resolution in amplitude and phase be degraded with the compressed ratio?

Response: We thank the reviewer for raising this question. The field-of-view (FOV) and the resolution of the holographic system operating in different modes are determined by the parameters of the two lenses in the 4f system and the physical size of the DMD. We added a section of “**Parameters in the large-FOV mode and the high-resolution mode**” in revised Supplementary Information to show the detailed analysis on evaluating these numbers.

Parameters in the large-FOV mode and the high-resolution mode

During experiments, we employed a square area with 768×768 effective pixels of the DMD. Since the pitch of the DMD is $19.35 \mu\text{m}$ (calculated above), the length of the diagonal lines along both the horizontal and vertical directions for the generated pattern are the same to be $19.35 \mu\text{m} \times 768 \approx 14.9 \text{ mm}$.

For the large-FOV mode demonstrated in this work, the two lenses employed for the $4f$ system are the same (AC254-125-A, Thorlabs). Thus, the surface of the DMD is 1:1 imaged to the surface of the sample. Considering a diffraction angle $\beta = 41.92^\circ$ that causes compression along the horizontal direction, the lengths of the diagonal lines along the horizontal and vertical directions are 14.9 mm and $14.9 \text{ mm} \times \cos(41.92^\circ) \approx 11.1 \text{ mm}$, respectively. Since a 3×3 pixels binning was employed, the corresponding resolution along the horizontal and vertical directions are $19.35 \mu\text{m} \times 3 \approx 58.0 \mu\text{m}$ and $19.35 \mu\text{m} \times \cos(41.92^\circ) \times 3 \approx 43.1 \mu\text{m}$, respectively. A larger FOV can be further achieved by amplifying the projected patterns using a different $4f$ system.

For the high-resolution mode demonstrated in this work, the two lenses employed for the $4f$ system have focal lengths of 300 mm (AC254-300-A, Thorlabs) and 30 mm (AC254-30-A, Thorlabs), respectively. Thus, the surface of the DMD is demagnified by a factor of 10 when imaged to the surface of the sample. In this case, both the FOV and the resolution are scaled down by a factor of 10 compared to that in the large-FOV mode. Thus, the FOV and the resolution along the horizontal and vertical directions are $1.49 \text{ mm} \times 1.11 \text{ mm}$ and $5.80 \mu\text{m} \times 4.31 \mu\text{m}$, respectively." A finer resolution can be further achieved by using a $4f$ system with a larger minification. However, such a $4f$ system requires special care due to the emergence of optical aberrations.

Besides, we also added descriptions on how the resolution degrades with sampling ratios in the Results section of the revised main text: Since the square path we employed for compressive sensing follows the order from low spatial frequency to high spatial frequency, the spatial resolution is expected to degrade along with the square root of the SR.

3. The authors claimed that the strategy of 768×768 pixels with 3×3 binning from DMD was performed in holographic imaging. Why not making full use of entire 768×1024 pixels with 3×4 binning? This strategy should provide a better SNR and imaging quality.

Response: We thank the reviewer for raising this question. We confirm that making full use of the entire 768×1024 pixels of the DMD can provide a better SNR indeed. However, such a choice will make the pattern on the surface of the DMD to be rectangular. Given a diffraction angle $\beta = 41.92^\circ$, the projected pattern on the sample turns into an asymmetric parallelogram, causing inconvenience for visualization purposes. Therefore, the effective area of the DMD is always kept as a square shape, so that the projected pattern on the surface of the sample exhibits a diamond shape. To clarify this point, we added the following sentence in the section of "**Operations of the digital micromirror device (DMD)**" in revised Supplementary Information: Due to this compression, a square pattern displayed by the DMD is transformed into a diamond shape. Notably, if a rectangular pattern is displayed by the DMD, this transformation turns the projected pattern into an asymmetric parallelogram shape. Thus, for visualization purposes, we restricted ourselves to use only square active areas of the DMD throughout this work.

4. In Fig. 2, lens L7 is regarded as a Fourier transformation of input field. When loading patterns with high spatial frequency, the output field is no longer a single point but exhibits spatial distribution. Considering the finite sense area of the single-pixel detector, it is possible that not all the output energy is collected. Are there any considerations or calculations to consider this condition?

Response: We thank the reviewer for raising an important question. As pointed out by the reviewer, when loading patterns with high spatial frequencies, the resulting field after passing through the collection lens is no longer a single point but exhibits spatial distribution. In this case, not all the output energy is collected if the sensing area of the single-pixel detector is not large enough. For single-pixel imaging, since the collected total energy becomes less for patterns with higher spatial frequencies, this phenomenon causes a degradation in image quality, especially for the fine details. Nonetheless, for single-pixel holography, a plane reference beam with zero spatial frequency is introduced. After passing through the collection lens, only the component with the same spatial frequency interferes between the reference beam and the signal beam. Since only the interference term is kept for image reconstruction, the single-pixel detector only needs to receive light with zero spatial frequency. As a result, the finite sensing area of the single-pixel detector does not influence the performance of single-pixel holography. This conclusion has been verified through numerical simulations in our previous studies (not shown in this work). To clarify this point, we added the following sentences in the Discussion section of the revised main text: In practice,

when loading patterns with high spatial frequencies, not all light can be collected by the photodetector if the sensing area is not large enough. This phenomenon can degrade the quality of the reconstructed images in SPI, especially for fine details. Nonetheless, it is surprising to find that SPH is immune to this effect. This observation is because only the interference term with zero spatial frequency contributes to the reconstruction process of SPH.

5. In the table on Page 3, the column of “refresh time” listed in this work is 48 μs , indicating that the full speed of the DMD (22k) was not fully used. Why not making full use of the 22k refresh rate to push the SBP-T to the limit?

Response: We thank the reviewer for this careful catch. The reason we chose a refresh time of 48 μs , i.e., a refresh rate of 20.8 kHz, is for the convenience of batch processing during imaging reconstruction. Given a sampling rate of 1.25 Ms/s of the data acquisition card and a beat frequency at 62,500 Hz, we can get 60 data points for the duration of one Hadamard pattern. Under this condition, 3 periods of heterodyne signals were measured and processed. In contrast, if we used the full speed of the DMD, we would get on average 56.8 data points for the duration of one Hadamard pattern. This choice only increases the data rate by 5% but causes considerable difficulties in batch processing data. To clarify this point, we added the following sentences in the Method section of the revised main text: “We note here that this DMD can support a shorter refresh time down to 45 μs (22 kHz refresh rate), corresponding to on average 56.8 data points for the duration of one Hadamard pattern. Although such a choice can gain an increase in data rate by roughly 5%, it causes difficulties in batch processing to reconstruct images.”

6. In the provided files “data_and_supporting_files.zip” and “readme.pdf”, it is mentioned that “To minimize phase drifting caused by environmental disturbance during the experiment.....”, could the authors provide a detailed explanation on the reasons of inserting additional Hadamard patterns and how to process these data corresponding to additional patterns?

Response: We thank the reviewer for catching this detail and we would like to clarify this point here. From the theoretical perspective, there is no need to insert such patterns. In practice, systems employing heterodyne holography are very sensitive to physical jitters and ambient noises, causing phase drifting during the data acquisition process. Such an effect can significantly deteriorate the quality of reconstructed holographic images. Moreover, based on Eq. (3) in revised main text that the determination of the complex coefficient for the n -th order Hadamard basis requires subtracting the measurement from the 1st order Hadamard pattern. Thus, we consider using the 1st order of Hadamard-like pattern for calibration naturally. Specifically, the 1st order of Hadamard-like pattern was inserted as a tracking base after displaying 16 different orders of Hadamard-like patterns. In this way, the drifted phase can be corrected over time by reference to the measurement of the 1st order of Hadamard-like pattern. We provided a detailed description to clarify this point in the section of “Heterodyne holography” in revised Supplementary Information: We note that the second term in Eq. (S6) is simply a constant value. Keeping this term during imaging reconstruction only imposes a constant background to the reconstructed images, which is acceptable for visualization. In practice, this term can be used to correct phase drifting that occurred during the data acquisition process. Specifically, in this work, the 1st order of Hadamard-like pattern was inserted as the tracking base after displaying every 16 orders of Hadamard-like patterns. The measurement of this recurring pattern was used to update the second term in Eq. (S6) over time.

7. In line 17 on Page 4, Eq(3), the first order of Hadamard-like patterns is simply a DC signal. My understanding is that if this value was not subtracted for each coefficient as the authors did, the final image would have a constant background. Nonetheless, for imaging purposes, a constant background is acceptable.

Response: We thank the reviewer for pointing this out. From the theoretical perspective, this term can be kept as the reviewer suggested. In practice, as shown in the response to Comment 6, this term was utilized as a tracking base to correct for the phase drifting that occurred during the data acquisition process. Detailed explanations are provided in revised Supplementary Information, which can be found in

the response to Comment 6.

8. What's the bandwidth of the photodiode? Specifications of this component are helpful for readers to adopt this technique.

Response: We thank the reviewer for this kind suggestion. We added the specification of the photodiode when describing the experimental setup in the Method section of the revised main text: ... **by a photodiode (DET10A2, Thorlabs) with a bandwidth of 350 MHz, which was then ...**

9. In Fig. 2, the DMD is not parallel to the sample plane. Will this introduce any measurement error or distortion?

Response: We thank the reviewer for asking this question. To make the illumination beam and the diffracted beam lie in the same horizontal plane, the rotational axis is perpendicular to the plane of the optical table. In this condition, the surface of the DMD is not orthogonal to neither the illuminating beam nor the diffracted beam to maximize the diffraction efficiency in this optical setup. The analysis of this choice has been analyzed and provided with details in the section of "**Operations of the digital micromirror device (DMD)**" in revised Supplementary Information. As a result, such an oblique configuration may induce a phase ramp across the projected image. Nonetheless, the calibration process we described in the section of "**The procedure of removing phase contaminations induced from the optical system**" can conveniently remove this affection. To clarify this issue, we added the following sentences in the section of "**Operations of the digital micromirror device (DMD)**" in revised Supplementary Information: "**This diffracted angle β will cause the displayed pattern not perpendicular to the propagation direction, inducing a phase ramp across the projected pattern. Nonetheless, the calibration process to correct phase contamination we described above can conveniently ease this affection. Moreover, this configuration also causes compression along the horizontal direction.**"

10. The gold standard is missing to cross-check the presented technique. Can the authors use existing techniques (e.g., phase-contrast microscopy) to verify their results? Also, what's the measurement accuracy? This would be another important parameter that would determine the application scope of this technique.

Response: We thank the reviewer for making this valuable suggestion. Unfortunately, we do not have a phase-contrast microscope in the lab. Meanwhile, the phase-contrast microscope cannot provide quantitative results to determine the accuracy of the measured phase values either. To overcome this weakness in the original submission, we performed new experiments by imaging a quantitative phase resolution target (QPT, BenchMark Tec) as a benchmark. The measurement accuracy in phase was quantified. We added the section of "**Holographic results of a quantitative phase resolution target**" in revised Supplementary Information, including both detailed descriptions and figures, to quantify the measurement accuracy of the holographic system in terms of phase.

Holographic results of a quantitative phase resolution target

To quantify the imaging capability in phase, a quantitative phase resolution target (QPT, BenchMark Tec) was imaged. This type of phase target is fabricated by coating transparent materials on a piece of glass. Specifically, phase patterns with the same size as groups 6-7 of the USAF standard resolution target are provided, allowing us to gauge the resolution in phase. While operating under the high-resolution mode, Fig. 6 shows the reconstructed amplitude and phase images of this phase target. Due to the low contrast in transmission, the amplitude image shown in Fig. 6(a) is vague. Nevertheless, the phase image shown in Fig. 6(b) exhibits great performance, manifesting delicate phase patterns that are almost blind in the amplitude counterpart. Specifically, we also quantified that the smallest structure that can be distinguished in phase is element 6 of group 6 (4.386- μm width), with a corresponding 1D profile displayed in the upper inset. This result agrees with the theoretical resolution under the high-resolution mode, which is 5.80 $\mu\text{m} \times 4.31 \mu\text{m}$.

Next, appraising whether the reconstructed phase value is quantitatively correct is another important issue. Given a coating thickness of 250 nm and a refractive index of 1.52, the phase difference between the phase pattern and the background is estimated to be $\Delta\varphi \approx 1.795$ rads for the green light. Here, we examined the bar located at the bottom of the phase target, enclosed in a dashed

rectangular. To minimize statistical errors, the same area was imaged four times. 1D profile that crosses both the bar and the background is depicted in the lower inset using blue circles, exhibiting a stepped structure. The errorbars represent the standard deviation of four independent measurements, resulting in an averaged fluctuation of about 0.137 rads. The averaged phase difference was estimated to be $\Delta\varphi_{\text{exp}} \approx 1.691$ rads, giving a phase error of only 0.104 rads ($\leq \lambda/60$). These results confirm that the developed SPH is quantitatively accurate to retrieve phase, showing prospects in biophotonics.

Fig. 6 Holographic results of imaging a quantitative phase resolution target under high-resolution mode. (a) The amplitude image of the quantitative phase resolution target, showing poor contrast. (b) The phase image of the quantitative phase resolution target, showing clear phase patterns. Upper inset: the corresponding one-dimensional profile of element 6 of group 6 (4.386- μm width). Lower inset: quantitative analysis on the accuracy of the reconstructed phase values, resulting in a phase error of about 0.104 rads. The corresponding scale bar is 100 μm .

Moreover, Fig. 7 shows the holographic results with compressive sensing. Similarly, when the SR decreases, the detailed structure gradually becomes obscure. Nonetheless, the holographic result reconstructed when $\text{SR} = 12.5\%$ is still acceptable.

Fig. 7 Holographic results of imaging the phase resolution target with compressive sensing. (a)(b) The amplitude and phase images reconstructed with different sampling ratios of 50%, 25%, 12.5%, 6.25%, and 3.125%. The corresponding scale bar is 200 μm .

Besides, we also make Fig. 6(b) as part of the newly added Fig. 5 in the Results section of the revised main text. Detailed descriptions on imaging a quantitative phase resolution target in the revised main text can be found in the Response to Comment 11.

11. In Fig. 5, the authors used red dashed circles to point out show some supplement information in the phase image from the amplitude counterpart. This part could be elaborated on because it showcases the capability of this technique.

Response: We thank the reviewer for this valuable suggestion. To showcase case the capability of this holographic technique that phase images can become a good supplement to the amplitude counterpart, we performed many new experiments on phase objects and unstained biological tissue. These holographic images confirmed that in many conditions, phase images can provide rich information with good contrast that is hardly seen in their amplitude counterparts. Detailed descriptions on imaging a quantitative phase resolution

target was added as a new section “**Holographic results of a quantitative phase resolution target**” in revised Supplementary Information, which can be found in the Response to Comment 10. Detailed descriptions on imaging unstained biological tissue were added as a new paragraph “**Holographic imaging of unstained tissue from mouse brains**” in the Results section of revised main text and a new section of “**Additional holographic images of unstained tissue from mouse brains**” in revised Supplementary Information.

In revised main text

Holographic imaging of unstained tissue from mouse brains. Having demonstrated holographic imaging of stained tissue from mouse tails, we then switched to image unstained tissue from mouse brains. Generally, images of unstained tissue exhibit low contrast in amplitude, but providing sufficient contrast through phase imaging. Before proceeding, we quantified the performance of the holographic system in phase by imaging a quantitative phase resolution target (QPT, BenchMark Tec). Figure 5(a) shows the reconstructed phase image of the resolution target, containing groups 6 and 7. To quantify the resolving power in phase, the corresponding 1D profile denoted by the black bracket within the resolution target, crossing element 6 of group 6 (4.386- μm width), is shown in the upper inset. Three verticle bars can be differentiated, indicating that the theoretical resolution under the high-resolution mode (5.80 $\mu\text{m} \times 4.31 \mu\text{m}$) was achieved in phase. We further quantified the accuracy of the measurement in phase by estimating the phase difference between the bar enclosed by a dashed box at the bottom of the target and its adjacent background. 1D profile that crosses both the bar and the background is shown in the lower inset using blue circles, exhibiting a stepped structure. The averaged phase difference was estimated to be $\Delta\varphi_{\text{exp}} \approx 1.691$ rads, yielding a phase error of only 0.104 rads ($\leq \lambda/60$) compared to the actual phase difference (1.795 rads). Detailed analysis on imaging the quantitative phase resolution target can be found in section VII of Supplementary Information. Then, we proceed to image unstained biological tissue. Figure 5(b) shows the image of an 80- μm -thick slice of unstained tissue from mouse brains, captured using a bright-field microscope. In this image, some representative types of tissue were denoted with black arrows, such as piriform cortex, perirhinal cortex, olivary pretectal nucleus, and white matter. For unstained tissue, it is usually challenging to identify different structures in amplitude, due to the insufficient contrast in transmission. Figures 5(c), (d), and (e) show a series of reconstructed holographic images for different parts of the mouse brain, indicated by three labeled diamond-shaped boxes in Fig. 5(b). Same as the one captured using the conventional microscope, these amplitude images only manifest rough outlines with low contrast. By comparison, phase images show rich details that can be hardly seen in their amplitude counterparts. For example, the piriform cortex and the perirhinal cortex are identifiable in phase images. Data used for producing Figs. 5(c)-(e) are provided in Supplementary Data 2. These results demonstrate the capability of employing SPH to image relatively transparent biological tissue.

Fig. 5 Performance of the high-throughput SPH with unstained tissue from mouse brains in high-resolution mode. (a) Reconstructed phase image of a quantitative phase target. The upper inset: the corresponding one-dimensional (1D) profile of element 6 of group 6; the lower inset: quantitative analysis on the accuracy of the reconstructed phase values, resulting in a phase error of 0.104 rads. (b) The image of a slice of unstained tissue from mouse brains, captured using a conventional microscope. Three diamond-shaped boxes represent the area being imaged by the holographic system. (c)-(e) The reconstructed amplitude and phase images for different parts of the unstained tissue. The corresponding scale bar is 200 μm.

For the selected area in Fig. 5(c), a series of holographic images reconstructed with compressive sensing at various SRs are illustrated in Fig. 6 as well. Similar to before, these images show the effectiveness of compressive sensing in dealing with biological tissue that contains rich information in phase. We also applied SPH to image other pieces of unstained tissues from mouse brains with different thicknesses, ranging from 10 μm to 120 μm, with details shown in section IX of Supplementary Information.

Fig. 6 Reconstruction of holographic images for the unstained tissue from mouse brains with compressive sensing. The amplitude and wrapped phase images are reconstructed with different sampling ratios of 50%, 25%, 12.5%, 6.25%, and 3.125%. The corresponding scale bar is 200 μm.

In revised Supplementary Information

Additional holographic images of unstained tissue from mouse brains

To show the imaging capability for unstained tissue, we also imaged a 100- μm -thick slice of unstained mouse brain. Figure 9(a) shows a bright-field image of this slice, captured using a conventional microscope. The imaged part of the tissue contains white matter and grey matter. Since the unstained slice is relatively thick with roughly the same transmission across the imaging region, it is challenging for the conventional microscope to provide good contrast. Figures 9(b), (c), and (d) show a series of reconstructed holographic images for different parts of the mouse brain, denoted by three labeled diamond-shaped boxes in Fig. 9(a). As expected, the features in amplitude images match well with that at corresponding areas in Fig. 9(a), which do not provide too much information with good contrast. As a comparison, phase images provide much better contrast, revealing many detailed structures that are indiscernible in their amplitude counterparts.

Fig. 9 Holographic performance of unstained mouse brain tissue with 100- μm thickness. (a) The image of a slice of 100- μm -thick unstained tissue from mouse brains, captured using a conventional microscope. Three diamond-shaped boxes represent the area being imaged by the holographic system. (b)-(d) The reconstructed amplitude and phase images for different parts of the unstained tissue. The corresponding scale bar is 200 μm .

We also imaged an even thicker slice of unstained mouse brain with a thickness of 120 μm . Figure 10(a) shows a bright-field image of this slice, captured using a conventional microscope. The imaged part contains the ventral part of the lateral septal nucleus, inferior colliculus, and white matter. Figures 10(b), (c), and (d) show a series of reconstructed holographic images for different parts of mouse brain, denoted by three labeled diamond-shaped boxes in Fig. 10(a). Again, phase images provide much better contrast, compared to their amplitude counterparts.

Fig. 10 Holographic performance of unstained mouse brain tissue with 120- μm thickness. (a) The image of a slice of 120- μm -thick unstained tissue from mouse brains, captured using a conventional microscope. Three diamond-shaped boxes represent the area being imaged by the holographic system. (b)-(d) The reconstructed amplitude and phase images for different parts of the stained tissue. The corresponding scale bar is 200 μm .

Very thin unstained tissue from mouse brains down to 10- μm thick was also imaged using our holographic system. Figure 11(a) shows a bright-field image of this slice, captured using a conventional microscope. The imaged part contains white matter and grey matter. Figures 11(b), (c), and (d) show a series of reconstructed holographic images for different parts of mouse brain, denoted by three labeled diamond-shaped boxes in Fig. 11(a). For such a thin tissue, it is hard to see details through the amplitude images. Nonetheless, phase images still provide rich information with good contrast.

Fig. 11 Holographic performance of unstained mouse brain tissue with 10- μm thickness. (a) The image of a slice of 10- μm -thick unstained tissue from mouse brains, captured using a conventional microscope. Three diamond-shaped boxes represent the area being imaged by the holographic system. (b)-(d) The reconstructed amplitude and phase images for different parts of the stained tissue. The corresponding scale bar is 200 μm .

12. A few typos are found in the manuscript. For example, by “mussel”, I believe the authors intended to say “muscle”.

Response: We thank the reviewer for catching it out. We carefully went through the revised submission and corrected such kinds of typos as many as we could.

Reviewer #3:

In this manuscript, a novel technique for single pixel compressive holography is proposed. The proposed technique employs heterodyne interference to achieve high-throughput measurement. The authors construct an optical system based on the proposed technique. And experimental verifications are described. In the experimental verifications, parts of a USAF chart for estimation of spatial resolution and biological tissues are reconstructed to demonstrate the usefulness of the system.

I have some questions and comments to improve the manuscript.

1. In Fig.2: Why the experimental set up consists of two AOMs? I think that heterodyne interferometry can be implemented with only one AOM with easy control. Also, the authors should describe the reason why the beat frequency is set to 62,500 Hz.

Response: We thank the reviewer for asking this important question. The AOM we used in the system is purchased from Intraaction company with a model number AOM-505AF1, which supports modulation frequencies in the range of 40 to 60 MHz. In this condition, if only one AOM was used, the beat frequency had to be set within the range of 40 to 60 MHz, which is far beyond the bandwidth of the data acquisition card we had (1.25Ms/s). Therefore, we chose to use the two AOMs to lower down the beat frequency to the range that the data acquisition card can handle. We clarified this point by adding the following information in the Method section of the revised main text: **...(AOM-505AF1, Intraaction, Optical frequency shift range: $\pm 40\sim 60$ MHz).** The reason why the beat frequency was set to 62,500 Hz is the compromise of the sampling rate of the data acquisition card (1.25Ms/s) and the refresh time of the DMD (48 μ s). The number of beating cycles is desired to be an integer for each displayed pattern. In this condition, a total number of 60 data points were measured for one displayed Hadamard pattern. For a beating frequency of 62,500 Hz, these 60 data points can be evenly distributed into 3 beating cycles. Indeed, there are many other choices for beating frequencies such as 125,000 Hz and 250,000 Hz, distributing these 60 data points into 6 and 12 beating cycles, respectively. Notably, for the same number of data points for one displayed Hadamard pattern, the quality of the reconstructed signal should not be sensitive on the choice of the beating frequency, provided the Nyquist sampling criterion was followed. Thus, we added the following sentence in the Method section of the revised main text: **What's more, the choice of the beating frequency is also not unique. In practice, there are many choices for beating frequencies such as 125,000 Hz and 250,000 Hz, distributing these 60 data points into 6 and 12 beating cycles, respectively. Notably, for the same number of data points for one displayed Hadamard pattern, the quality of the reconstructed signal should not be sensitive on the choice of the beating frequency, provided the Nyquist sampling criterion was followed. An integer number of beating cycles for each displayed pattern is also desired for computational convenience.**

2. About Fig. 4 (a): the authors should discuss factors that the image quality at sampling ratio 50% is worse than that in more critical case (25% & 12.5%). The discussions are important for judgement which the used algorithm for image reconstruction is. Such Discussion are important in determining whether the algorithm used for image reconstruction is practical.

Response: We thank the reviewer for this valuable suggestion. In general, Hadamard-like patterns with higher orders contain higher spatial frequency, so that information carried by these orders is more sensitive to measurement noises. Thus, although compressive sensing with a smaller sampling ratio may lose fine details in the reconstructed image, it is more robust to measurement noises. This is the reason why the image quality at a sampling ratio of 50% is worse than that at sampling ratios of 25% and 12.5%. As suggested by the reviewer, we added a few sentences in the Results section of the revised main text to explain this observation and added the section of **"Investigations on compressive sensing under different noise levels"** in revised Supplementary Information to quantitatively study this effect.

In revised main text:

We note that this observation is because information carried by Hadamard-like patterns with higher orders is more sensitive to measurement noises. A quantitative study on how measurement noises affect the image quality of the reconstructed image under different choices of SRs can be found in section V of Supplementary Information. Thus, the SR of 25% might be a suitable choice for the noise

level in current experimental conditions (the standard deviation of the measurement noise is around 0.1% of the measured value).

In revised Supplementary Information:

“Investigations on compressive sensing under different noise levels

In practice, measurement noises contributed from shot noises of light and electronic noises of equipment, causing degradation in image quality. Using numerical tools, here, we quantify how this effect impacts the performance of compressive sensing with different sampling ratios (SRs).

The original holographic image S was set with a resolution of 256×256 . We adopted a direct inverse basis transformation with a square path for compressive sensing, as described in main text. For simplicity, measurement noises were simulated as white Gaussian noises with various standard deviations [1]. In particular, noise levels ranging from 0 to 1% of the averaged measurement values were considered. For compressive sensing, various SRs at 3.125%, 6.25%, 12.5%, 25%, 50%, and 100% were employed to reconstruct images O_{CS} under different noise levels. A first-order correlation function between the reconstructed and original images, defined as $|(O_{CS}, S)| / (|(O_{CS}, O_{CS})|(S, S))^{1/2}$, was used to quantify the similarities between these two holographic images.

Figure 3(a) plots correlations as a function of noise levels when holographic images were reconstructed at different SRs. For all cases, the correlations decrease as the noises become large. However, correlations correspond to different SRs decay at different speeds. In general, the larger the SR is, the faster the correlation decays. Thus, although the reconstruction process with a small SR performs worse for small measurement noises, it turns out to be better when measurement noises become large. This observation indicates that compressive sensing with a smaller SR, which concentrates more on the lower spatial frequency, is more robust to measurement noises. In our experiment, the noise level is around 0.1% of the measurement, denoted as a yellow area in the figure. To identify which SR is the most suitable one under the current experimental condition, we then fixed the noise level of 0.1% in the following simulation. Under such conditions, blue circles in Fig. 3(b) describe the decreased correlation solely induced by the noise, while red circles describe the evolution of the correlation purely resulted from the changing of SRs in compressive sensing. Notably, the trends of these two effects indicate that there exists a suitable SR under the current noise level. By multiplying these two effects, yellow circles represent the combined effect. One could see that the maximum correlation is achieved at SR = 25%. This result explains why the reconstructed images with SR = 100% and 50% look worse than those with SR = 25% and 12.5% in main text. Nonetheless, we emphasize here that although a large SR is susceptible to measurement noises, it still provides a high-resolution image, as the decreased correlation is mainly contributed from the noisy background but not from the fine feature.

Fig. 3 Simulation results on compressive sensing under different noise levels. (a) Correlations between the reconstructed and original images as a function of different noise levels. Compressive sensing realized with different sampling ratios (SRs) were investigated, which are labeled with different colors. The yellow area represents the actual noise levels during experiments. (b) After fixing the noise level as 0.1% of the measurement value, the SR-resultant effect and the noise-induced effect are represented using blue and red circles as a function of the SR. The combined effect is represented using yellow circles, exhibiting a maximum value at SR = 25%.

3. About holographic imaging: To show the effectiveness as holographic imaging, phase objects or three-dimensional objects should be experimentally verified. This is because verification for phase imaging with the constructed experimental setup is not enough. (On the other hands, that for amplitude reconstruction is quite enough to show the usefulness.)

Response: We thank the reviewer for raising this valuable question. To show the effectiveness of holographic imaging, we supplemented a series of new experiments to image a quantitative phase resolution target (Benchmark) and several slices of unstained biological tissue from mouse brains. These holographic images show rich information in phase that is hardly discernible in the amplitude counterpart.

We added the section of “**Holographic results of a quantitative phase resolution target**” in revised Supplementary Information, including both detailed descriptions and figures, to quantify the measurement accuracy of the holographic system in phase values.

Holographic results of a quantitative phase resolution target

To quantify the imaging capability in phase, a quantitative phase resolution target (QPT, BenchMark Tec) was imaged. This type of phase target is fabricated by coating transparent materials on a piece of glass. Specifically, phase patterns with the same size as groups 6-7 of the USAF standard resolution target are provided, allowing us to gauge the resolution in phase. While operating under the high-resolution mode, Fig. 6 shows the reconstructed amplitude and phase images of this phase target. Due to the low contrast in transmission, the amplitude image shown in Fig. 6(a) is vague. Nevertheless, the phase image shown in Fig. 6(b) exhibits great performance, manifesting delicate phase patterns that are almost blind in the amplitude counterpart. Specifically, we also quantified that the smallest structure that can be distinguished in phase is element 6 of group 6 (4.386- μm width), with a corresponding 1D profile displayed in the upper inset. This result agrees with the theoretical resolution under the high-resolution mode, which is $5.80 \mu\text{m} \times 4.31 \mu\text{m}$.

Next, appraising whether the reconstructed phase value is quantitatively correct is another important issue. Given a coating thickness of 250 nm and a refractive index of 1.52, the phase difference between the phase pattern and the background is estimated to be $\Delta\varphi \approx 1.795$ rads for the green light. Here, we examined the bar located at the bottom of the phase target, enclosed in a dashed rectangular. To minimize statistical errors, the same area was imaged four times. 1D profile that crosses both the bar and the background is depicted in the lower inset using blue circles, exhibiting a stepped structure. The errorbars represent the standard deviation of four independent measurements, resulting in an averaged fluctuation of about 0.137 rads. The averaged phase difference was estimated to be $\Delta\varphi_{\text{exp}} \approx 1.691$ rads, giving a phase error of only 0.104 rads ($\leq \lambda/60$). These results confirm that the developed SPH is quantitatively accurate to retrieve phase, showing prospects in biophotonics.

Fig. 6 Holographic results of imaging a quantitative phase resolution target under high-resolution mode. (a) The amplitude image of the quantitative phase resolution target, showing poor contrast. (b) The phase image of the quantitative phase resolution target, showing clear phase patterns. Upper inset: the corresponding one-dimensional profile of element 6 of group 6 (4.386- μm width). Lower inset: quantitative analysis on the accuracy of the reconstructed phase values, resulting in a phase error of about 0.104 rads. The corresponding scale bar is 100 μm .

Moreover, Fig. 7 shows the holographic results with compressive sensing. Similarly, when the SR decreases, the detailed structure gradually becomes obscure. Nonetheless, the holographic result reconstructed when $\text{SR} = 12.5\%$ is still acceptable.

Fig. 7 Holographic results of imaging the phase resolution target with compressive sensing. (a)(b) The amplitude and phase images reconstructed with different sampling ratios of 50%, 25%, 12.5%, 6.25%, and 3.125%. The corresponding scale bar is 200 μm .

Detailed descriptions on imaging unstained biological tissue were added as a new paragraph “**Holographic imaging of unstained tissue from mouse brains**” in the Results section of revised main text and a section of “**Additional holographic images of unstained tissue from mouse brains**” in revised Supplementary Information.

In revised main text

Holographic imaging of unstained tissue from mouse brains. Having demonstrated holographic imaging of stained tissue from mouse tails, we then switched to image unstained tissue from mouse brains. Generally, images of unstained tissue exhibit low contrast in amplitude, but providing sufficient contrast through phase imaging. Before proceeding, we quantified the performance of the holographic system in phase by imaging a quantitative phase resolution target (QPT, BenchMark Tec). Figure 5(a) shows the reconstructed phase image of the resolution target, containing groups 6 and 7. To quantify the resolving power in phase, the corresponding 1D profile denoted by the black bracket within the resolution target, crossing element 6 of group 6 (4.386- μm width), is shown in the upper inset. Three verticle bars can be differentiated, indicating that the theoretical resolution under the high-resolution mode (5.80 $\mu\text{m} \times 4.31 \mu\text{m}$) was achieved in phase. We further quantified the accuracy of the measurement in phase by estimating the phase difference between the bar enclosed by a dashed box at the bottom of the target and its adjacent background. 1D profile that crosses both the bar and the background is shown in the lower inset using blue circles, exhibiting a stepped structure. The averaged phase difference was estimated to be $\Delta\varphi_{\text{exp}} \approx 1.691$ rads, yielding a phase error of only 0.104 rads ($\leq \lambda/60$) compared to the actual phase difference (1.795 rads). Detailed analysis on imaging the quantitative phase resolution target can be found in section VII of Supplementary Information. Then, we proceed to image unstained biological tissue. Figure 5(b) shows the image of an 80- μm -thick slice of unstained tissue from mouse brains, captured using a bright-field microscope. In this image, some representative types of tissue were denoted with black arrows, such as piriform cortex, perirhinal cortex, olivary pretectal nucleus, and white matter. For unstained tissue, it is usually challenging to identify different structures in amplitude, due to the insufficient contrast in transmission. Figures 5(c), (d), and (e) show a series of reconstructed holographic images for different parts of the mouse brain, indicated by three labeled diamond-shaped boxes in Fig. 5(b). Same as the one captured using the conventional microscope, these amplitude images only manifest rough outlines with low contrast. By comparison, phase images show rich details that can be hardly seen in their amplitude counterparts. For example, the piriform cortex and the perirhinal cortex are identifiable in phase images. Data used for producing Figs. 5(c)-(e) are provided in Supplementary Data 2. These results demonstrate the capability of employing SPH to image relatively transparent biological tissue.

Fig. 5 Performance of the high-throughput SPH with unstained tissue from mouse brains in high-resolution mode. (a) Reconstructed phase image of a quantitative phase target. The upper inset: the corresponding one-dimensional (1D) profile of element 6 of group 6; the lower inset: quantitative analysis on the accuracy of the reconstructed phase values, resulting in a phase error of 0.104 rads. (b) The image of a slice of unstained tissue from mouse brains, captured using a conventional microscope. Three diamond-shaped boxes represent the area being imaged by the holographic system. (c)-(e) The reconstructed amplitude and phase images for different parts of the unstained tissue. The corresponding scale bar is 200 μm .

For the selected area in Fig. 5(c), a series of holographic images reconstructed with compressive sensing at various SRs are illustrated in Fig. 6 as well. Similar to before, these images show the effectiveness of compressive sensing in dealing with biological tissue that contains rich information in phase. We also applied SPH to image other pieces of unstained tissues from mouse brains with different thicknesses, ranging from 10 μm to 120 μm , with details shown in section IX of Supplementary Information.

Fig. 6 Reconstruction of holographic images for the unstained tissue from mouse brains with compressive sensing. The amplitude and wrapped phase images are reconstructed with different sampling ratios of 50%, 25%, 12.5%, 6.25%, and 3.125%. The corresponding scale bar is 200 μm .

Additional holographic images of unstained tissue from mouse brains

To show the imaging capability for unstained tissue, we also imaged a 100- μm -thick slice of unstained mouse brain. Figure 9(a) shows a bright-field image of this slice, captured using a conventional microscope. The imaged part of the tissue contains white matter and grey matter. Since the unstained slice is relatively thick with roughly the same transmission across the imaging region, it is challenging for the conventional microscope to provide good contrast. Figures 9(b), (c), and (d) show a series of reconstructed holographic images for different parts of the mouse brain, denoted by three labeled diamond-shaped boxes in Fig. 9(a). As expected, the features in amplitude images match well with that at corresponding areas in Fig. 9(a), which do not provide too much information with good contrast. As a comparison, phase images provide much better contrast, revealing many detailed structures that are indiscernible in their amplitude counterparts.

Fig. 9 Holographic performance of unstained mouse brain tissue with 100- μm thickness. (a) The image of a slice of 100- μm -thick unstained tissue from mouse brains, captured using a conventional microscope. Three diamond-shaped boxes represent the area being imaged by the holographic system. (b)-(d) The reconstructed amplitude and phase images for different parts of the unstained tissue. The corresponding scale bar is 200 μm .

We also imaged an even thicker slice of unstained mouse brain with a thickness of 120 μm . Figure 10(a) shows a bright-field image of this slice, captured using a conventional microscope. The imaged part contains the ventral part of the lateral septal nucleus, inferior colliculus, and white matter. Figures 10(b), (c), and (d) show a series of reconstructed holographic images for different parts of mouse brain, denoted by three labeled diamond-shaped boxes in Fig. 10(a). Again, phase images provide much better contrast, compared to their amplitude counterparts.

Fig. 10 Holographic performance of unstained mouse brain tissue with 120- μm thickness. (a) The image of a slice of 120- μm -thick unstained tissue from mouse brains, captured using a conventional microscope. Three diamond-shaped boxes represent the area being imaged by the holographic system. (b)-(d) The reconstructed amplitude and phase images for different parts of the stained tissue. The corresponding scale bar is 200 μm .

Very thin unstained tissue from mouse brains down to 10- μm thick was also imaged using our holographic system. Figure 11(a) shows a bright-field image of this slice, captured using a conventional microscope. The imaged part contains white matter and grey matter. Figures 11(b), (c), and (d) show a series of reconstructed holographic images for different parts of mouse brain, denoted by three labeled diamond-shaped boxes in Fig. 11(a). For such a thin tissue, it is hard to see details through the amplitude images. Nonetheless, phase images still provide rich information with good contrast.

Fig. 11 Holographic performance of unstained mouse brain tissue with 10- μm thickness. (a) The image of a slice of 10- μm -thick unstained tissue from mouse brains, captured using a conventional microscope. Three diamond-shaped boxes represent the area being imaged by the holographic system. (b)-(d) The reconstructed amplitude and phase images for different parts of the stained tissue. The corresponding scale bar is 200 μm .

4. In conclusion: some performances of the proposed method are shown. Those estimations are much important in the paper. The authors should summarize the performance as a table like Table 1 and discuss on relations between scalability and specifications of the elemental devices in detail.

Response: We thank the reviewer for this valuable suggestion. In the Method section of the revised main text, we summarized the performance of three operational modes in Tab. 2 and discussed the relations between scalability and specifications in detail: **As a final remark, we briefly compare the performance of several operational modes developed in this work, which are summarized in Tab. 2. Two large-FOV modes and one high-resolution mode were demonstrated with the same illumination device, beat frequency, and the sampling rate of DAC. Thus, the same SBP- T was achieved for all three operational modes, calculated by multiplying the reciprocal of refresh time of the illumination device by 2. Moreover, the two large-FOV modes share the same $4f$ system, while the only difference is the binning strategy. Due to this reason, the lateral resolution and the FOV of these two modes are simply scaled by a factor of 1.5. As for the high-resolution mode, a different $4f$ system was used. Therefore, the lateral resolution and the FOV of this mode are scaled to be ten times smaller than that in the first large-FOV mode. Detailed evaluations on these parameters can be found in section X of Supplementary Information. Currently, the switching between the large FOV mode and the high-resolution mode requires hardware modification of the $4f$ optical setup, which is inconvenient and prevents real-time operation. Nonetheless, we anticipate that a variable $4f$ setup that consists of tunable lenses and an advanced electromechanics system with auto-focusing can enable smooth adjustment between different operational modes in the future.**

Table. 2 List of the parameters in both the large-FOV mode and the high-resolution mode.

		Large-FOV mode 1 (Main Text)	Large-FOV mode 2 (Supplementary Information)	High-resolution mode 1 (Main Text & Supplementary Information)
Illumination device	Pixel size (μm)	13.68		
	Refresh time (ms)	0.048		
Beating frequency (Hz)		62,500		
Sampling rate of DAC (Ms/s)		1.25		
Strategy of binning pixels		768 \times 768 pixels with 3 \times 3 binning	512 \times 512 pixels with 2 \times 2 binning	768 \times 768 pixels with 3 \times 3 binning
Lens in $4f$ system	front model	AC254-125-A, Thorlabs (125mm)	AC254-125-A, Thorlabs (125mm)	AC254-300-A, Thorlabs (300mm)
	rear model	AC254-125-A, Thorlabs (125mm)	AC254-125-A, Thorlabs (125mm)	AC254-030-A, Thorlabs (30mm)
Lateral resolution (μm)		58.0 \times 43.1	38.7 \times 28.8	5.80 \times 4.31
FOV (mm)		14.9 \times 11.1	9.91 \times 7.37	1.49 \times 1.11
SBP- T (pixels/s)		41,666.6		

5. Forms of reconstructed images in Figs. 3, 4, 5, and 6: (This comment may be not so essential.) Why area of measurement is not square like but diamond shape (rhomboid)?

Response: We thank the reviewer for raising a good question. To make the illumination beam and the diffracted beam lie in the same horizontal plane, the rotational axis is perpendicular to the plane of the optical table. Thus, the effective area of the DMD, i.e., a square area, should be rotated in plane by 45° . Moreover, to maximize the diffraction efficiency of the two-dimensional blazed grating, a diffraction angle $\beta = 41.92^\circ$ was chosen. This diffraction compresses the dimension along the horizontal direction, transforming the projected pattern from a square shape to a diamond shape. Detailed analysis can be found in the section of “**Operations of the digital micromirror device (DMD)**” in revised Supplementary Information. To clarify this issue and avoid confusion, we added the following sentences at the end of the section: **Due to this compression, a square pattern displayed by the DMD is transformed into a diamond shape. Notably, if a rectangular pattern is displayed by the DMD, this transformation turns the projected pattern into an asymmetric parallelogram shape. Thus, for visualization purposes, we restricted ourselves to use only square active areas of the DMD throughout this work.**

Reviewers' Comments:

Reviewer #1:

Remarks to the Author:

The authors present a revised version of the optical heterodyne detection scheme for single-pixel holographic imaging along the lines raised by the different reviewers.

The main scientific message of the manuscript has now been emphasized and clarity of the manuscript has improved so that is easy to follow. The study extends current knowledge about single-pixel holography.

Technical data have been improved and new experiments have been performed. Laboratory experiments are now at the heart of the manuscript and have been rationally designed to support the claims made in the paper. Also, previously published material have been cited in accordance with reviewers' suggestions.

Thus, I recommend acceptance of the manuscript in Nature Communications.

Reviewer #2:

Remarks to the Author:

In this revised manuscript, the authors have addressed all my comments. It is always good to see the authors spent time directly answering the comments with new data. I have to admit that I am impressed by the author's diligence in performing many additional experiments, which have considerably improved the quality of an already excellent article. In these new results, the imaging of unstained mouse brain tissue makes the technique clearly stand out from existing methods (e.g., HE histology). In this regard, the proposed method, thanks to its scalable FOV and relatively high speed, has the promising potential of becoming a generic tool for many biomedical applications. Thus, I recommend publish it as is.

Reviewer #3:

Remarks to the Author:

Dear authors,

In this revised version of the manuscript and supplementary informations, the authors are considered to respod to my questions and comments clearly.

The revised set of the manuscript is much improved in comparison with the original one.

Point-by-point responses to reviewers' comments

Reviewer #1:

The authors present a revised version of the optical heterodyne detection scheme for single-pixel holographic imaging along the lines raised by the different reviewers.

The main scientific message of the manuscript has now been emphasized and clarity of the manuscript has improved so that is easy to follow. The study extends current knowledge about single-pixel holography.

Technical data have been improved and new experiments have been performed. Laboratory experiments are now at the heart of the manuscript and have been rationally designed to support the claims made in the paper. Also, previously published material have been cited in accordance with reviewers' suggestions.

Thus, I recommend acceptance of the manuscript in Nature Communications.

Response: We are glad that our reply addresses the reviewer's question and also thank the reviewer for recognizing our work. The reviewer's unique insight on the reconstructive algorithm about single-pixel imaging enables a deeper comprehension in the realm of compressive sensing and adaptive smart sensing. Some representative references the reviewer listed also broaden our horizon about real-time demonstrations in single-pixel imaging through deep learning.

Reviewer #2:

In this revised manuscript, the authors have addressed all my comments. It is always good to see the authors spent time directly answering the comments with new data. I have to admit that I am impressed by the author's diligence in performing many additional experiments, which have considerably improved the quality of an already excellent article. In these new results, the imaging of unstained mouse brain tissue makes the technique clearly stand out from existing methods (e.g., HE histology). In this regard, the proposed method, thanks to its scalable FOV and relatively high speed, has the promising potential of becoming a generic tool for many biomedical applications. Thus, I recommend publish it as is.

Response: We are happy that the revised submission addresses all the reviewer's comments and also thank the reviewer for recognizing our work. The reviewer provides lots of detailed comments, such as verification of the correctness of phase imaging and the clarification of certain choices of parameters during experiments. These comments are critical and help us improve the demonstration of the imaging capability of our holographic system.

Reviewer #3:

Dear authors,

In this revised version of the manuscript and supplementary information, the authors are considered to respond to my questions and comments clearly.

The revised set of the manuscript is much improved in comparison with the original one.

Regards,

Response: We are grateful that the revised submission satisfactorily responds to the questions and comments of the reviewer and thank the reviewer for recognizing our work. The reviewer's suggestion is very helpful, enabling us to improve the capability of the holographic system by adding experiments on phase resolution targets and relatively transparent samples. The reviewer's comments also urge us to consider compressive sensing under different noise levels.